# FedHera: Towards Drift-Resilient Federated Fine-tuning with Heterogeneous Resources

**Ke Xiao**[1]  **Qiyuan Wang**[1]  **Christos Anagnostopoulos**[1]  **Zhuoran Tan**[1]  **Wenhao Li**[1]

## Abstract

Driven by the imperative to leverage privacy-sensitive data scattered across decentralized devices, federated fine-tuning has emerged as a vital paradigm for adapting large language models without compromising data privacy. Yet, its practical efficacy is bottlenecked by severe client resource heterogeneity. Existing truncation-based methods typically couple the transmitted rank with the trainable rank, which (i) under-utilizes bandwidth on communication-rich but compute-limited clients and (ii) exacerbates truncation-induced gradient drift. To address this, we propose FEDHERA, a resource-decoupled framework that explicitly differentiates information reception from gradient optimization. FEDHERA employs a spectrum-preserving allocation strategy to maximize the transfer of global knowledge (via high-rank singular values) within bandwidth limits, irrespective of training constraints. Furthermore, we introduce a prefix-gating mechanism that utilizes the downloaded high-capacity basis as a frozen reference to guide local updates, thereby minimizing the optimization gap caused by aggressive truncation. Extensive experiments under different heterogeneous settings show that FEDHERA improves stability and accuracy over state-of-the-art baselines.

## 1. Introduction

Parameter-efficient fine-tuning (PEFT) techniques, such as Low-Rank Adaptation (LoRA) (Hu et al., 2022), have significantly lowered the barrier to adapting large language models (LLMs) by reducing memory and optimization costs (Zhang et al., 2026a). However, effective adaptation relies not only on efficient algorithms but also on high-quality domain data, which is often inherently decentralized and privacy-sensitive. While Federated Learning (FL) offers a principled framework to leverage such decentralized data without compromising privacy, directly applying it to LLMs via full-parameter training imposes prohibitive communication and memory overheads. To bridge this gap, Federated Fine-Tuning (FFT) integrates FL with PEFT (Singhal et al., 2025b; Peng et al., 2024), enabling clients to collaboratively adapt LLMs by exchanging compact adapter updates rather than raw data or full model parameters.

Despite its potential, deploying FFT at scale is impeded by the diverse communication and computation capabilities of participating devices. In LoRA-based frameworks, the adapter rank $r$ serves as the primary lever for balancing model capacity against resource consumption. To address this, recent heterogeneous frameworks, such as FlexLoRA (Bai et al., 2024) and FedHL (Peng et al., 2025), introduce rank-adaptive mechanisms, yet they typically couple the downloaded rank to the client's trainable rank, enforcing $r^{\text{train}} \equiv r^{\text{tot}}$. These methods conflate communication bandwidth with computational memory. Consider a typical edge scenario involving devices like NVIDIA Jetson or high-end smartphones. These devices often possess high-speed connectivity (e.g., WiFi 6 or 5G) capable of streaming large model parameters, yet are strictly bottlenecked by limited VRAM which prohibits the gradient computation for high-rank matrices. As a result, forcing the download rank to drop to the level of the training rank discards valuable global information, exacerbating truncation-induced drift.

To break this bottleneck, we propose FEDHERA (**Fed**erated **He**terogeneous **R**ank **A**daptation), a drift-resilient heterogeneous FFT framework that decouples the downloaded inference rank from the local trainable rank to reconcile communication bandwidth with local computational constraints. Unlike prior coupled approaches, FEDHERA allows the downloaded inference rank to exceed the trainable rank ($r^{\text{tot}} > r^{\text{train}}$), thereby converting excess bandwidth into improved feature fidelity. Central to our design is a dual-rank mechanism: utilizing a spectrum-preserving water-filling strategy to maximize the global basis reception, and a prefix-gating protocol that trains only a budget-feasible prefix. This

---

[1]School of Computing Science, University of Glasgow, Glasgow, United Kingdom. Correspondence to: Ke Xiao <ke.xiao@glasgow.ac.uk>.

*Proceedings of the 43rd International Conference on Machine Learning*, Seoul, South Korea. PMLR 306, 2026. Copyright 2026 by the author(s).

design converts excess bandwidth into improved feature fidelity by utilizing the unoptimized frozen tail as a stable spectral anchor, effectively mitigating the optimization drift inherent in aggressive truncation.

We evaluate FEDHERA on a diverse suite of reasoning and generative tasks. In scenarios with highly skewed resource distributions, FEDHERA demonstrates consistent gains over SOTA baselines. These empirical results validate that utilizing a richer global subspace via the frozen tail effectively anchors local updates to the global trajectory, thereby securing both drift resilience and superior generalization.

Our main contributions are:

- We identify the cost asymmetry between parameter transmission and gradient computation as an unexploited opportunity. We show that decoupling these constraints allows extra frozen components to serve as spectral anchors, thereby mitigating the optimization drift exacerbated by rigid coupling.

- We propose FEDHERA, a resource-decoupled framework that separates information reception from gradient optimization. By maintaining the downloaded-but-untrained tail active in the forward pass and progressively integrating it, FedHera effectively mitigates truncation-induced local drift.

- Extensive experiments across three model families (Jiang et al., 2023; Touvron et al., 2023; Grattafiori et al., 2024) demonstrate consistent improvements over state-of-the-art baselines (Bai et al., 2024; Peng et al., 2025; Zhang et al., 2025) in drift resilience and generalization.

## 2. Related Work

### 2.1. Federated Fine-tuning

The paradigm of collaborative learning has shifted from training from scratch to FFT of foundation models. While initial frameworks (Ye et al., 2024; Zhang et al., 2024) and full-parameter studies (Qin et al., 2024) explored this frontier, the prohibitive communication overhead of LLMs has firmly established PEFT as the standard. Methods like SLoRA (Babakniya et al., 2023) and zeroth-order optimization (Ling et al., 2024) have demonstrated that efficient, gradient-light adaptation is viable in edge scenarios.

However, a persistent challenge in distributed LoRA lies in the instability and weight divergence arising from independent local updates and the separate aggregation of low-rank factors. To mitigate these issues, recent efforts have introduced stabilizing mechanisms, such as residual-style aggregation corrections (Singhal et al., 2025a; Yan et al., 2025), fixing one projection matrix to enhance optimization

stability (Sun et al., 2024), or selective factor sharing to balance global and client-specific knowledge (Guo et al., 2025). Together with client-customized adaptation schemes (Kim et al., 2023), these approaches improve the robustness and personalization of federated LoRA. Nevertheless, they primarily address optimization stability, aggregation bias, or statistical heterogeneity, rather than the system-level question of how adapter capacity should be allocated when clients differ in communication and computation resources.

### 2.2. Heterogeneous Federated Fine-tuning

Real-world federated deployments face significant heterogeneity across both data distributions and hardware resources. While data heterogeneity has been explored through data-efficient instruction tuning (Qin et al., 2025) and dynamic task allocation (Bai et al., 2024; Wu et al., 2025), system heterogeneity imposes a more rigid constraint for LoRA-based federated fine-tuning. Standard LoRA typically assumes a uniform adapter rank across clients, effectively capping the global model capacity at the weakest participant (Kuang et al., 2024).

To relax this bottleneck, recent methods tailor adaptation capacity to client resource budgets. Rank-heterogeneous and SVD-based approaches (Cho et al., 2023; Wang et al., 2024; Bai et al., 2024) allocate different ranks to different clients, while FedHL (Peng et al., 2025) further mitigates aggregation bias under heterogeneous low-rank updates. Other strategies explore structural heterogeneity, such as heterogeneous layer allocation in FedHeLLo (Zhang et al., 2025), parallel one-rank modules in Fed-PLoRA (Zhang et al., 2026b), and multi-head low-rank reparameterization in Ravan (Raje et al., 2026).

Despite these advances, most existing methods still assign each client a single effective adapter capacity, tying the global information a client receives to the capacity it can locally train and upload. This rigid coupling overlooks a practical asymmetry: clients may have sufficient bandwidth and static memory to receive a richer global basis, but insufficient optimizer-state memory or compute budget to train all of it. Consequently, informative global components can be discarded before local training, increasing truncation-induced drift and weakening cross-client alignment. FEDHERA addresses this gap by decoupling information reception from gradient optimization under heterogeneous communication and computation constraints.

## 3. Preliminaries and Motivation

### 3.1. SVD-based Aggregation for Federated LoRA

FFT typically leverages LoRA to circumvent the prohibitive costs of full-parameter training. By freezing the pre-trained weights $\mathbf{W}_0$ and optimizing low-rank adapters

$\mathbf{B} \in \mathbb{R}^{d \times r}$ and $\mathbf{A} \in \mathbb{R}^{r \times p}$, LoRA learns a low-rank update $\Delta \mathbf{W} = s\,\mathbf{BA}$ and applies it to the frozen weights: $\mathbf{W} = \mathbf{W}_0 + \Delta \mathbf{W}$, where $s$ is a scalar scaling factor. However, applying standard FedAvg directly to these decoupled matrices introduces a mathematical inconsistency known as aggregation bias: the summation of products is not equivalent to the product of summations, i.e.,

$$\sum_{i \in \mathcal{N}} \mathbf{B}_i \mathbf{A}_i \neq (\sum_{i \in \mathcal{N}} \mathbf{B}_i)(\sum_{i \in \mathcal{N}} \mathbf{A}_i). \tag{1}$$

Standard aggregation mandates homogeneous ranks across all clients, as seen in benchmarks like (Kuang et al., 2024), failing to accommodate system heterogeneity. To resolve these issues, recent SVD-based approaches propose aggregating the reconstructed updates $\mathbf{W}_g = \sum p_i \mathbf{B}_i \mathbf{A}_i$, where $p_i$ is the aggregation weight of client $i$. The server performs SVD on the global update $\mathbf{W}_g$:

$$\mathbf{W}_g = \mathbf{U}\boldsymbol{\Sigma}\mathbf{V}^T. \tag{2}$$

This decomposition establishes a unified global basis. To support heterogeneous clients, the server truncates these matrices to a client-specific rank $r_i$, deriving the personalized adapters as:

$$\mathbf{B}_i = \mathbf{U}[:, : r_i]\boldsymbol{\Sigma}_{:r_i}^{1/2}, \quad \mathbf{A}_i = \boldsymbol{\Sigma}_{:r_i}^{1/2}\mathbf{V}[:, : r_i]^T. \tag{3}$$

This formulation allows the server to extract and distribute the principal components corresponding to a specific rank $r_i$, thereby tailoring the adapter size to each client's capacity.

### 3.2. Challenges in Heterogeneous Federated Fine-tuning

While SVD-based methods enable rank heterogeneity, they inherit a rigidly coupled resource allocation strategy where the communication and inference ranks are strictly bound to the local training rank (Peng et al., 2025; Bai et al., 2024). This imposes two critical limitations.

**Inefficiency of Coupled Resource Allocation.** This design overlooks a significant cost asymmetry between storing static parameters and maintaining dynamic training states. Since active training requires typically 4 to 6 times more memory for optimizer states than static weights (Rajbhandari et al., 2021), clients often possess sufficient bandwidth to receive high-rank adapters but lack the VRAM to train them. Rigid coupling forces these clients to downscale their download rank to match the training bottleneck, leading to underutilized communication resources and the loss of global information cheap to load for inference.

**Truncation-Induced Optimization Drift.** Standard heterogeneity-aware methods introduce a significant optimization gap by initializing clients in a heavily truncated subspace (Bai et al., 2024). Theoretical studies indicate this deviation accumulates over epochs, driven by the initial

truncation error (Peng et al., 2025). By forcing the inference rank to match the bottlenecked training rank, existing methods incur large initialization errors that inevitably lead to severe gradient drift. Consequently, stabilizing convergence requires minimizing the initialization error independently of training constraints.

## 4. The FEDHERA Framework

### 4.1. Overview

FEDHERA is founded on the principle of resource decoupling. Departing from prior approaches that conflate transmission and optimization constraints, we explicitly treat them as distinct budgets to maximize information utilization. Central to our framework is a unified global basis derived via server-side SVD. We leverage this basis to implement a dual-rank allocation Strategy: maximizing the downloaded inference rank $r^{\text{tot}}$ to preserve global feature fidelity, while restricting the trainable rank $r^{\text{train}}$ to fit local memory limits.

As illustrated in Figure 1, the framework operates in a coordinated closed loop. In the server phase, we employ a spectrum-preserving water-filling algorithm to dynamically assign the $(r^{\text{tot}}, r^{\text{train}})$ pair for each layer, strictly prioritizing principal components with higher spectral energy, as detailed in Section 4.2. In the client phase, clients execute local fine-tuning via a prefix-gating mechanism, as elaborated in Section 4.3. This technique leverages the 'frozen tail' of the downloaded basis as a stable reference to guide the optimization of the active prefix, thereby mitigating the gradient drift inherent in heterogeneous truncation.

### 4.2. Resource-Aware Heterogeneous Rank Allocation

We formulate the rank allocation task as a constrained optimization problem. For each client $i$ and layer $\ell$, the server assigns a dual-rank configuration $\{r_{i,\ell}^{\text{tot}}, r_{i,\ell}^{\text{train}}\}$ subject to $0 \leq r_{i,\ell}^{\text{train}} \leq r_{i,\ell}^{\text{tot}}$. The objective is to maximize the total retained spectral energy across the model within client-specific communication ($B_i$) and memory ($M_i$) budgets. To solve this, we propose a Layer-wise Spectrum-Preserving Water-Filling scheme. Unlike uniform allocation strategies, it dynamically distributes rank budgets based on layer-wise spectral sensitivity by assigning higher ranks to layers with slowly decaying singular values, thereby minimizing reconstruction error.

**Spectral Energy Quantification.** Following the aggregation of client updates, the server performs SVD on the global update, ensuring singular values are sorted in descending order. By the Eckart–Young–Mirsky theorem (Eckart & Young, 1936), the optimal low-rank approximation error is minimized by retaining the largest singular values. We

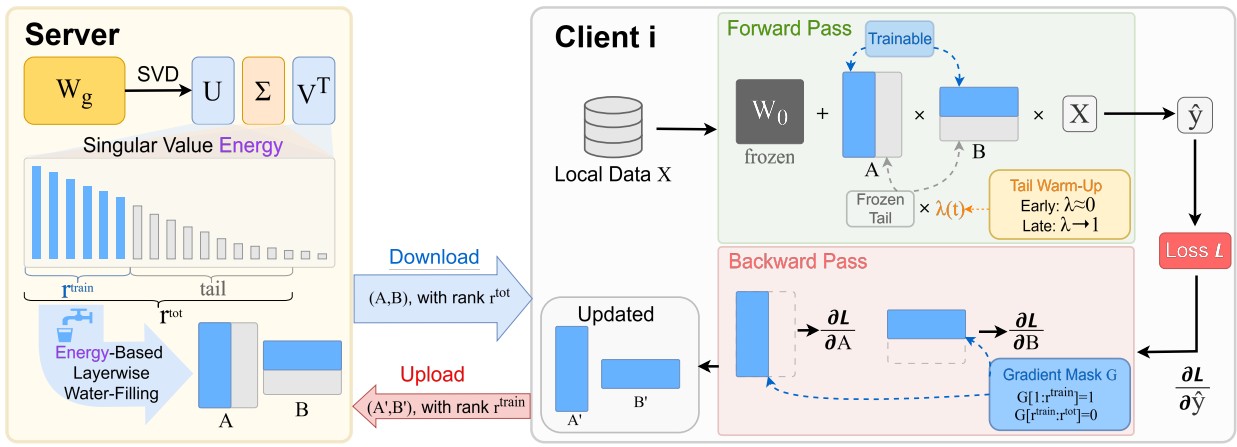

*Figure 1.* **Overview of FEDHERA.** The server maintains a unified global basis via SVD and utilizes energy-aware water-filling to decouple the download rank $r^{\text{tot}}$ from the trainable rank $r^{\text{train}}$. Clients employ a prefix-gating mechanism to update only the trainable prefix while leveraging the frozen tail with an adaptive gate $\lambda_i^{(t)}$ to mitigate drift.

quantify the relative importance of the $k$-th singular value $\sigma_{\ell,k}$ in layer $\ell$ using a layer-normalized energy:

$$\mathcal{E}_{\ell,k} = \frac{\sigma_{\ell,k}^2}{\sum_j \sigma_{\ell,j}^2 + \epsilon}. \quad (4)$$

$\epsilon > 0$ is introduced for numerical stability. This normalization yields a scale-normalized spectral score that is comparable across layers with different parameter magnitudes. The allocator therefore prioritizes components with the largest marginal retained energy per unit resource cost, rather than raw singular values alone.

**Water-Filling Strategy.** We formulate the rank allocation as a variant of the Knapsack Problem, solved via a greedy water-filling algorithm to maximize retained spectral energy under strict resource budgets. We define the allocation efficiency of the $k$-th singular value in layer $\ell$ as:

$$\eta_{\ell,k} = \frac{\mathcal{E}_{\ell,k}}{\mathcal{C}_{\ell,k}}, \quad (5)$$

where $\mathcal{C}_{\ell,k}$ denotes the marginal resource cost. We apply this unified logic to dual constraints by deriving distinct cost functions:

- **Download Allocation:** To maximize global information reception within the downlink bandwidth budget $B_i$, the marginal cost is defined as the transmission overhead per rank:

$$\mathcal{C}_{i,\ell}^{\text{down}} = (d_{\ell,\text{out}} + d_{\ell,\text{in}}) \cdot b_i, \quad (6)$$

where $d$ denotes layer dimensions and $b_i$ represents the bytes per parameter. The algorithm iteratively increments $r_{i,\ell}^{\text{tot}}$ for indices with the highest efficiency $\eta$ until the budget $B_i$ is exhausted (Alg. 1).

---

**Algorithm 1** Water-Filling for $r_{i,\ell}^{\text{tot}}$ Rank Allocation

---

**Input**: Client $i$'s downlink budget $B_i$; per-layer costs $c_{i,\ell}^{\text{down}}$; layer-normalized energies $\mathcal{E}_{i,\ell,k}$
**Output**: $r_{i,\ell}^{\text{tot}}$ for all layers $\ell$

1: Initialize residual budget $\widehat{B} \leftarrow B_i$; ranks $r_{i,\ell}^{\text{tot}} \leftarrow 0, \forall \ell$
2: **while** $\widehat{B} \geq \min_\ell c_{i,\ell}^{\text{down}}$ **do**
3: $\quad \ell^* \leftarrow \arg\max_\ell \{\mathcal{E}_{\ell,r_{i,\ell}^{\text{tot}}+1} / c_{i,\ell}^{\text{down}} : r_{i,\ell}^{\text{tot}} < R_\ell\}$
4: $\quad$ **if** $\ell^* = \text{null} \,||\, \widehat{B} < c_{i,\ell^*}^{\text{down}}$ **then**
5: $\quad\quad$ **break**
6: $\quad$ **end if**
7: $\quad r_{i,\ell^*}^{\text{tot}} \leftarrow r_{i,\ell^*}^{\text{tot}} + 1, \widehat{B} \leftarrow \widehat{B} - c_{i,\ell^*}^{\text{down}}$
8: **end while**
9: **return** $r_{i,\ell}^{\text{tot}}$

---

- **Training Allocation:** To constrain local optimization within memory $M_i$ and latency $T_i$ limits, we construct a composite computational cost. The resource cost per rank is defined by its marginal memory $c_{i,\ell}^{\text{mem}}$ and computation time $c_{i,\ell}^{\text{time}}$, which are unified via adaptive scalarization:

$$\mathcal{C}_{i,\ell}^{\text{train}} = \alpha \cdot c_{i,\ell}^{\text{time}} + \beta \cdot c_{i,\ell}^{\text{mem}}, \quad (7)$$

the marginal costs $c_{i,\ell}^{\text{mem}}$ and $c_{i,\ell}^{\text{time}}$ are obtained via a lightweight bootstrap profile under the target training configuration, while $\alpha$ and $\beta$ are updated online from the residual resource budgets rather than manually tuned. Crucially, the optimization of $r_{i,\ell}^{\text{train}}$ is performed within the subspace defined by the download allocation, strictly enforcing the nesting constraint $r_{i,\ell}^{\text{train}} \leq r_{i,\ell}^{\text{tot}}$ (Alg. 2).

**Algorithm 2** Water-Filling for $r_{i,\ell}^{\text{train}}$ Rank Allocation

---

**Input**: Client $i$'s time and memory budgets $T_i, M_i$; per-layer caps $r_{i,\ell}^{\text{tot}}$; per-layer time/memory slopes $c_{i,\ell}^{\text{time}}, c_{i,\ell}^{\text{mem}}$; layer-normalized energies $\mathcal{E}_{\ell,k}$

**Output**: $r_{i,\ell}^{\text{train}}$ for all layers $\ell$

1: Initialize residual budgets $\widehat{T} \leftarrow T_i, \widehat{M} \leftarrow M_i - M_{\text{static}}$
2: Initialize $r_{i,\ell}^{\text{train}} \leftarrow 0, \forall \ell$
3: **while** $\exists \ell : (r_{i,\ell}^{\text{train}} < r_{i,\ell}^{\text{tot}} \wedge c_{i,\ell}^{\text{time}} \leq \widehat{T} \wedge c_{i,\ell}^{\text{mem}} \leq \widehat{M})$ **do**
4:    $\tilde{T} \leftarrow 1/\max(\widehat{T}/T_i, \epsilon), \quad \tilde{M} \leftarrow 1/\max(\widehat{M}/M_i, \epsilon)$
5:    $\alpha \leftarrow \tilde{T}/(\tilde{T} + \tilde{M}), \quad \beta \leftarrow \tilde{M}/(\tilde{T} + \tilde{M})$
6:    $\ell^* \leftarrow \arg\max_\ell \{\mathcal{E}_{\ell, r_{i,\ell}^{\text{train}}+1}/(\alpha\, c_{i,\ell}^{\text{time}} + \beta\, c_{i,\ell}^{\text{mem}}) : r_{i,\ell}^{\text{tot}} < R_\ell \wedge (c_{i,\ell}^{\text{time}}, c_{i,\ell}^{\text{mem}}) <_{\text{pw}} (\widehat{T}, \widehat{M})\}$
7:    **if** $\ell^* = $ null **then**
8:       **break**
9:    **end if**
10:   $r_{i,\ell^*}^{\text{train}} \leftarrow r_{i,\ell^*}^{\text{train}} + 1$
11:   $\widehat{T} \leftarrow \widehat{T} - c_{i,\ell^*}^{\text{time}}, \quad \widehat{M} \leftarrow \widehat{M} - c_{i,\ell^*}^{\text{mem}}$
12: **end while**
13: **return** $r_{i,\ell}^{\text{train}}$

---

### 4.3. Prefix-Gated Training and Unbiased Aggregation

Given the allocated dual-rank configuration $\{r_{i,\ell}^{\text{tot}}, r_{i,\ell}^{\text{train}}\}$, the server initializes the client-specific low-rank adapters. By performing SVD on the global weight $\mathbf{W}_g$, we construct the downlink adapters $\mathbf{B}_i \in \mathbb{R}^{d \times r^{\text{tot}}}$ and $\mathbf{A}_i \in \mathbb{R}^{r^{\text{tot}} \times k}$ utilizing the top-$r^{\text{tot}}$ singular components. This initialization bridges the gap between transmission efficiency and feature fidelity, setting the stage for client optimization.

**Prefix-Gated Local Training.** To strictly enforce local memory constraints while retaining high-fidelity global contexts, FEDHERA adopts a prefix-gating mechanism. The local optimizer is restricted to update only the first $r_{i,\ell}^{\text{train}}$ ranks, while the remaining $r_{i,\ell}^{\text{tot}} - r_{i,\ell}^{\text{train}}$ ranks serve as a frozen spectral buffer (the frozen tail). We apply a binary mask $\mathbf{G}_{i,\ell}$ to the gradient flow during the backward pass:

$$\nabla_{\mathbf{A}_i}\mathcal{L} \leftarrow \nabla_{\mathbf{A}_i}\mathcal{L} \odot \mathbf{G}_{i,\ell}, \tag{8}$$

$$\mathbf{G}_{i,\ell} = \text{diag}(\underbrace{1,\ldots,1}_{r_{i,\ell}^{\text{train}}}, \underbrace{0,\ldots,0}_{r_{i,\ell}^{\text{tot}} - r_{i,\ell}^{\text{train}}}). \tag{9}$$

Although the optimizer states are instantiated only for the trainable prefix, the forward pass utilizes the full $r^{\text{tot}}$-rank subspace. The frozen tail acts as a stable reference anchor, ensuring the local feature distribution remains aligned with the global manifold, thereby implicitly mitigating the gradient drift caused by heterogeneous truncation.

**Server-Side:** Upon training completion in round $t$, clients transmit only the updated parameters of the active prefix $\mathbf{B}_i^{\text{up}}, \mathbf{A}_i^{\text{up}}$, ensuring uplink communication scales efficiently

with $\mathcal{O}(r^{\text{train}})$. To reconstruct the global model, we depart from standard parameter averaging, which suffers from dimensionality mismatch in heterogeneous settings. Instead, we leverage the residual aggregation strategy proposed in (Peng et al., 2025). The server updates the full-rank global weights $\mathbf{W}_g^{(t+1)}$ by accumulating the weighted residuals:

$$\mathbf{W}_g^{(t+1)} = \mathbf{W}_g^{(t)} + \sum_{i \in \mathcal{N}} p_i \left(\mathbf{B}_i^{\text{up}}\mathbf{A}_i^{\text{up}} - \mathbf{B}_i^{\text{init}}\mathbf{A}_i^{\text{init}}\right), \tag{10}$$

where superscripts *up* and *init* denote updated and initial states, respectively. By subtracting the initial term, Equation (10) ensures that the singular components truncated during the downlink phase are preserved losslessly in $\mathbf{W}_g^{(t)}$. This guarantees that the global update trajectory is driven solely by valid client gradients, eliminating the systematic bias introduced by our heterogeneous rank allocation.

Note that Eq. (10) is a server-side remedy for aggregation bias under rank-heterogeneous updates, and we adopt it following (Peng et al., 2025) as a stabilization plug-in. Our main focus is different: FEDHERA targets client-side optimization drift induced by aggressive truncation and rigid resource coupling. The dual-rank design and the frozen-tail forward anchoring with ATW shape the local optimization trajectory, which is orthogonal to the choice of residual vs. standard aggregation and can be combined with either.

### 4.4. Adaptive Tail Warm-up

While the dual-rank design increases adapter capacity, the reliability of the global basis is inherently time-dependent. In early communication rounds, the global basis is estimated from sparse and highly heterogeneous client updates, so its tail components may encode high-variance noise or client-specific artifacts rather than stable, shared directions. Over-activating these immature tail directions as a fixed prior can bias the optimization of the trainable prefix, leading to unstable or slowed convergence.

To address this, we propose Adaptive Tail Warm-up (ATW), which treats the frozen tail as a progressive prior and modulates its contribution based on the alignment between a client's local update and the aggregated global update, increasing the tail influence only as the basis becomes reliable.

**Server-Side:** Upon aggregating updates, the server quantifies the reliability of the global update direction relative to each client $i$. We compute a directional alignment score $s_i^{(t)}$ using the cosine similarity of the update matrices under the Frobenius inner product:

$$s_i^{(t)} = \frac{\langle \Delta\mathbf{W}_i^{up}, \Delta\mathbf{W}_g \rangle_F}{\|\Delta\mathbf{W}_i^{up}\|_F \|\Delta\mathbf{W}_g\|_F}, \tag{11}$$

where $\Delta\mathbf{W}_i^{up} = \mathbf{B}_i^{\text{up}}\mathbf{A}_i^{\text{up}}$ represents the local update submitted by client $i$, and $\Delta\mathbf{W}_g = \sum p_k \Delta\mathbf{W}_k$ denotes the

aggregated global update for the current round. This score drives the update of the next round gating factor:

$$\lambda_i^{(t+1)} = 1 - \exp\left(-\frac{t}{2} \cdot (1 + s_i^{(t)} \cdot \beta^{t-\hat{t}})\right). \quad (12)$$

Here $\beta^{t-\hat{t}}$ serves as a staleness penalty for clients last active at round $\hat{t}$. Equation (12) ensures the influence of the frozen tail grows asymptotically from 0 to 1, accelerating only when the global direction aligns consistently with local updates.

**Client-Side:** During local fine-tuning, the client incorporates the gating factor into the forward pass, modulating the contribution of the frozen tail:

$$\mathbf{h} = \mathbf{W}_0\mathbf{x} + \mathbf{B}^{\text{train}}\mathbf{A}^{\text{train}}\mathbf{x} + \lambda_i \cdot \mathbf{B}^{\text{tail}}\mathbf{A}^{\text{tail}}\mathbf{x}. \quad (13)$$

Initializing $\lambda \approx 0$ suppresses the tail prior in early rounds and gradually activates it as global basis becomes reliable.

The architectural logic of FEDHERA transforms fundamental resource constraints into operational advantages. This prioritization strategy is theoretically grounded in our analysis of heterogeneous truncation error in Appendix A. When communication is relatively ample, FEDHERA can provision a richer global basis via a larger $r^{\text{tot}}$ while still keeping $r^{\text{train}}$ within the client's optimization budget, and the proposed ATW further prevents immature tail directions from destabilizing early-round training.[1]

# 5. Experiments

We evaluate FEDHERA on a diverse suite of language tasks under heterogeneous client resources.[2] We report main results on performance and generalization across uniform and skewed resource distributions, and compare against SOTA heterogeneous baselines. To complement these end-to-end results, we analyze optimization dynamics in Section 6, including truncation-induced drift and the role of prefix-gating in mitigating it.

## 5.1. Experimental Setup

**Models and Datasets.** We evaluate generalization across three task categories: (1) **Arithmetic Reasoning: Mistral-7B** fine-tuned on GSM8K and MetaMathQA; (2) **Commonsense Reasoning: LLaMA-2 (7B)** on HellaSwag, BoolQ, PIQA, and WinoGrande; (3) **Generative Tasks: Llama-3.2 (3B)** on E2E NLG and Alpaca. Dataset statistics and preprocessing details are provided in Appendix C.

---

[1]Even when $r^{\text{tot}} = r^{\text{train}}$, FEDHERA still differs from coupled baselines by employing energy-aware, layer-wise rank allocation to retain more spectral energy than uniform-rank assignment. This prioritization is supported by the ablation in Table 12.

[2]Our code is available at https://github.com/shock-xiaoke/FedHera.

**Federated Simulation and Baselines.** We simulate federated fine-tuning under two resource-heterogeneity profiles: a **Uniform** distribution and a **Skewed** distribution that includes resource outliers to mimic realistic deployments. We compare FEDHERA against a conventional homogeneous-rank baseline, denoted **FedHomoLoRA** for clarity, where all clients are constrained to the minimum rank budget. We further compare against SOTA heterogeneous methods: rank-adaptive approaches such as **FlexLoRA** (Bai et al., 2024) and **FedHL** (Peng et al., 2025), as well as pruning-based methods such as **FedHeLLo** (Zhang et al., 2025).

**Implementation Details.** Unless otherwise stated, we employ residual-based unbiased aggregation in Section 4.3 for improved stability. We simulate $N = 100$ clients and sample a fixed fraction of clients per communication round. This setting follows common practice in federated fine-tuning studies and enables controlled evaluation under heterogeneous budgets. Full hyperparameter configurations are listed in Appendix E.

## 5.2. Main Results

**Evaluation Protocol.** To assess cross-client generalization, we follow a held-out, pre-update evaluation protocol used in prior heterogeneous FFT studies (Bai et al., 2024). At the start of each round, we evaluate the aggregated global model on the local validation sets of clients not selected for training in that round. We report the average improvement over a homogeneous-rank baseline under matched local training budgets.

**Reasoning Tasks.** We begin with six arithmetic and commonsense reasoning benchmarks. Figure 2 summarizes accuracy gains under the Skewed resource distribution.

**Results.** These results directly address the two challenges raised in Section 3: (i) coupled allocation wastes available bandwidth on compute-limited clients, and (ii) rank truncation induces optimization drift that harms convergence. FEDHERA consistently achieves the strongest gains across tasks by allowing clients to receive a higher-rank global basis ($r^{\text{tot}}$) while optimizing only a budget-feasible prefix ($r^{\text{train}}$), improving forward fidelity without increasing optimizer-state footprint. The improved generalization is aligned with the reduced truncation-induced deviation analyzed in Section 6. Uniform results are reported in Appendix F.1, and final-round accuracy is in Appendix F.2.

**Generative Tasks.** We evaluate generation on two settings: instruction tuning on Alpaca and data-to-text generation on E2E NLG. We use ROUGE-L as the primary metric, and additionally report evaluation loss (negative log-likelihood) under the same held-out, pre-update protocol to complement overlap-based scores with a token-level

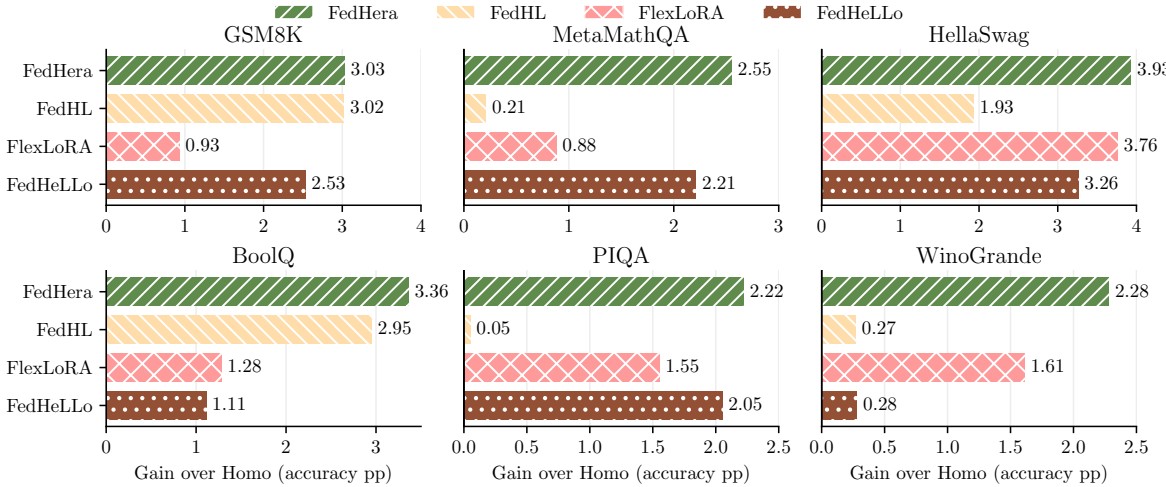

*Figure 2.* **Performance gain on reasoning benchmarks (Skewed distribution).** Accuracy improvement (percentage points) over the homogeneous baseline across six reasoning tasks. FEDHERA consistently outperforms coupled heterogeneity-aware baselines. (Uniform distribution results are detailed in Appendix F.1.)

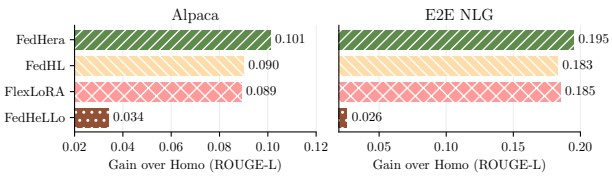

*Figure 3.* **Performance gain on generative tasks (Skewed distribution).** ROUGE-L improvement (absolute gain) over the homogeneous baseline on Alpaca and E2E NLG datasets.

*Table 1.* Evaluation loss on generative tasks under the *Skew.* partition, evaluated with the held-out, pre-update protocol.

| METHOD | ALPACA ↓ | E2E NLG ↓ |
|---|---|---|
| FLEXLORA | 1.194 | 0.530 |
| FEDHL | 1.186 | 0.513 |
| FEDHELLO | 1.617 | 0.834 |
| **FEDHERA** | **1.140** | **0.492** |

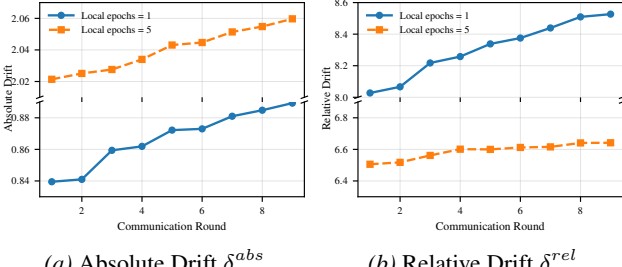

*(a) Absolute Drift $\delta^{abs}$*  *(b) Relative Drift $\delta^{rel}$*

*Figure 4.* **Evolution of Drift Metrics.** We monitor the drift of FEDHERA over communication rounds with varying local epochs $E \in \{1, 5\}$ (Qwen3-0.6B on E2E NLG).

*Table 2.* Comparison of trajectory drift $\delta^{abs}$ and $\delta^{rel}$ on the E2E NLG task using Qwen3-0.6B.

| METHOD | ABS. DRIFT ↓ | REL. DRIFT ↓ |
|---|---|---|
| FLEXLORA | 2.991 | 9.647 |
| FEDHL | 2.072 | 6.684 |
| FEDHELLO | 4.170 | 12.223 |
| **FEDHERA** | **2.060** | **6.641** |

likelihood measure.

**Results.** Figure 3 shows ROUGE-L gains under Skewed distribution, where FEDHERA achieves the largest improvements on both datasets. Table 1 further shows the lowest evaluation loss, indicating that the gains are not limited to overlap metrics but also reflect improved token-level likelihood. Overall, these gains align with FEDHERA's objective of reducing truncation-induced drift while leveraging communication–optimization asymmetry, which we further validate through the drift and efficiency analyses in Section 6. Additional metrics (BLEU, NIST, METEOR, and CIDEr) are provided in Appendix F.2.

## 6. Analysis

We analyze FEDHERA from a mechanism-oriented perspective to explain where its gains come from. Specifically, we assess: (i) do federated low-rank constraints distort client updates relative to an ideal local optimizer, and can FED-HERA reduce this drift? (ii) does our energy-aware rank allocation contribute beyond a naive allocation? and (iii) does FEDHERA improve memory saturation and information throughput under heterogeneous budgets, compared to coupled baselines? Unless otherwise stated, all analyses are conducted under the Skewed distribution.

### 6.1. Drift from Ideal Local Updates

A key motivation of FEDHERA is to suppress the directional deviation introduced by low-rank truncation during local training. To quantify this effect, we compare the actual federated local update with an *oracle* update.

**Setup and Metrics.** For each client $i$, we obtain an oracle update $\Delta\mathbf{W}_i^\star$ by fine-tuning a high-rank adapter ($r = 1024$), which approximates the update direction while avoiding full-parameter training. We then measure the discrepancy (Frobenius distance) between the federated update $\Delta\mathbf{W}_i^{up}$ and $\Delta\mathbf{W}_i^\star$ using the absolute drift:

$$\delta_i^{abs} = ||\Delta\mathbf{W}_i^{up} - \Delta\mathbf{W}_i^\star||_F, \tag{14}$$

and the relative drift:

$$\delta_i^{rel} = \frac{||\Delta\mathbf{W}_i^{up} - \Delta\mathbf{W}_i^\star||_F}{||\Delta\mathbf{W}_i^\star||_F + \epsilon}, \tag{15}$$

where $\epsilon > 0$ is a small constant for numerical stability.

**Results and Interpretation.** Table 2 reports final-round drift on E2E NLG with Qwen3-0.6B. FlexLoRA and Fed-HeLLo exhibit substantially larger drift, suggesting that tightly coupling the communicated rank with the trainable rank amplifies truncation-induced projection mismatch. In contrast, FEDHERA achieves the lowest absolute and relative drift. Figure 4 further shows the drift evolution across rounds. Notably, increasing local epochs from 1 to 5 reduces $\delta_i^{rel}$, while $\delta_i^{abs}$ increases due to larger update magnitudes, indicating improved directional alignment with the oracle update. To stress-test stability under prolonged local training, we vary $E \in \{1, 3, 5, 7\}$ in Fig. 5. Coupled baselines exhibit clear error compounding as local steps increase, whereas FEDHERA suppresses the drift growth rate, consistent with the frozen-tail acting as a forward-pass anchor to the global principal subspace.

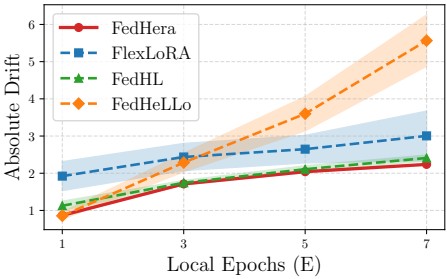

*Figure 5.* **Accumulation of Drift over Local Epochs.** Drift accumulation as a function of local epochs $E \in \{1, 3, 5, 7\}$.

Beyond quantifying optimization stability, we empirically validate that our drift metrics serve as faithful predictors of downstream generalization quality. Focusing on the high-drift scenario $E = 5$, we report the final average ROUGE-L

scores across all clients: FEDHERA (0.7922) > FEDHL (0.7866) > FLEXLORA (0.7768) > FEDHELLO (0.6961). This performance hierarchy is perfectly inversely correlated with the magnitude of the measured drift, where FEDHERA exhibited the lowest $\delta^{abs}$ and FEDHELLO the highest. This alignment suggests that $\delta^{abs}/\delta^{rel}$ are not merely theoretical statistics, but indicative proxies for the robustness of federated optimization. While we acknowledge this relationship is correlational, it strongly supports the hypothesis that mitigating projection-induced drift directly translates into superior generative performance.

We further assess cross-client drift by reconstructing each participating client's residual update in a common dense parameter space and measuring round-wise directional dispersion as $1 - \cos(\text{vec}(\Delta W_i^{up}), \text{vec}(\Delta W_j^{up}))$. FEDHERA exhibits consistently lower dispersion across communication rounds, indicating that decoupled rank reception not only reduces deviation from a high-rank local reference, but also provides a stronger shared spectral anchor for aligning heterogeneous client updates. Detailed curves and layer-wise analyses are provided in Appendix F.11.

### 6.2. Further Ablations and Efficiency Diagnostics

Beyond the primary benchmarks, we provide additional diagnostics to (i) isolate the contribution of key design choices and (ii) quantify the practical efficiency benefits under heterogeneous budgets. Detailed results are deferred to Appendix F.

**Rank Allocation.** We validate the spectral Water-Filling allocator against uniform and random alternatives. Results confirm that prioritizing high-energy components yields superior fidelity under matched budgets (Appendix F.5).

**System-Level Efficiency.** We evaluate memory saturation and information throughput. FEDHERA achieves 98.7% VRAM utilization and a $2.27\times$ information gain ratio, significantly outperforming coupled baselines (Appendix F.7).

**Robustness and Components.** We report robustness under non-IID data skew, showing consistent gains across varying $\alpha$ (Appendix F.3). We also verify that FEDHERA retains superiority over baselines even when reverting to standard aggregation, confirming that our gains stem primarily from the dual-rank mechanism rather than server-side corrections (Appendix F.6). Finally, component ablations verify the necessity of adapting all linear modules and the stabilizing role of ATW in early rounds (Appendix F.9, F.8).

## 7. Conclusion

We proposed FEDHERA, a drift-resilient federated fine-tuning framework that addresses system heterogeneity by decoupling the downloaded inference rank from the local train-

able rank. FEDHERA maximizes global information reception via an energy-aware, layer-wise water-filling allocator, while prefix-gated training (with Adaptive Tail Warm-up) updates only a budget-feasible subspace and uses the frozen tail as a forward-pass anchor to mitigate truncation-induced drift. Across three model families and diverse reasoning and generative benchmarks under uniform and skewed resource profiles, FEDHERA consistently improves accuracy, generation quality, and stability over state-of-the-art heterogeneous baselines, while achieving higher resource utilization. These results suggest that resource decoupling is a practical and principled design axis for scalable federated adaptation of foundation models.

## Acknowledgements

This work was supported in part by the EU Horizon projects TERRA (Grant Agreement No. 101189962) and ELLIE (Grant Agreement No. 101178099). We also thank the reviewers for their thoughtful comments and constructive suggestions, which helped strengthen the paper.

## Impact Statement

This paper aims to contribute to fundamental advances in machine learning methodology. While such progress can have a wide range of societal impacts, we do not identify any specific, unusual, or direct implications arising from this work beyond those typically associated with advances in this field.

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

# A. Theoretical Analysis of Truncation Error

In this section, rather than deriving a standard convergence rate which relies heavily on convex assumptions often violated by LLMs, we focus on quantifying the approximation error induced by our heterogeneous rank allocation and its implications for generalization. Specifically, we derive a bound on the expected risk difference between the full-rank global model and the client-side heterogeneous low-rank approximations. To make the comparison precise, we follow the analytical setup and work under the standard assumptions adopted in FlexLoRA and related studies (Malladi et al., 2023; Hua et al., 2023).

We use $f(\cdot; x, y)$ to denote the per-sample loss and $h(\cdot; x)$ to denote the hypothesis (model output) for input $x$. The expectation of the loss on dataset $\mathcal{D}$ is denoted as $\mathcal{L}(W) \triangleq \mathbb{E}_{(x,y) \sim \mathcal{D}} f(W; (x, y))$.

## A.1. Assumptions and Lemmas

**Assumption A.1** (Smoothness of LoRA parameters). There exist constants $L_f > 0$ and $L_h > 0$ such that for any LoRA weight matrices $W, W'$ and any sample $(x, y)$, the following Lipschitz continuity conditions hold:

$$\left| f(W; x, y) - f(W'; x, y) \right| \leq L_f \|W - W'\|_F, \tag{16}$$

$$\|h(W; x) - h(W'; x)\|_2 \leq L_h \|W - W'\|_F. \tag{17}$$

**Assumption A.2** (Bounded LoRA weights and truncation error). There exists $R > 0$ such that any global LoRA weight matrix $W_g$ satisfies $\|W_g\|_F \leq R$. Let $\text{SVD}(W_g, r^i)$ denote the best rank-$r^i$ approximation of $W_g$ obtained by keeping the top $r^i$ singular values. We assume for the aggregated $W_g$, there exists a constant $\phi > 0$ such that the truncation error is bounded:

$$\left\| \text{SVD}(W_g, r^i) - W_g \right\|_F \leq \phi^i. \tag{18}$$

Before presenting the main theorem, we introduce the following lemma regarding the optimality of SVD truncation, which is central to our analysis of FedHera.

**Lemma A.3** (Optimality of Truncated SVD). *Consider a matrix $W \in \mathbb{R}^{d \times p}$ and let $q = \min(d, p)$. Let $\sigma_1 \geq \sigma_2 \geq \cdots \geq \sigma_q \geq 0$ denote the singular values of $W$. According to the Eckart–Young–Mirsky theorem (Eckart & Young, 1936), for any integer $r$ such that $1 \leq r < q$, the optimal truncated SVD approximation $W^{(r)}$ satisfies:*

$$\|W - W^{(r)}\|_F^2 = \min_{A: rank(A) \leq r} \|W - A\|_F^2 = \sum_{j=r+1}^{q} \sigma_j^2. \tag{19}$$

*This implies that minimizing the Frobenius norm of the approximation error is equivalent to minimizing the sum of the squared energies of the discarded singular values.*

## A.2. Main Convergence Theorem

**Theorem A.4.** *Under Assumptions A.1 and A.2, with probability at least $1 - \delta$, there exists a sample size threshold $\tilde{N}_{Hera}$ such that for all global weights $W_g$, the generalization error bound $\|\mathcal{L}(W_g) - \mathcal{L}(W_g')\| \leq \epsilon$ holds when the number of local data samples for each client $i$ exceeds $\tilde{N}_{Hera}$, where:*

$$\tilde{N}_{Hera} = \mathcal{O}\left( \frac{k}{|\mathcal{N}|\epsilon^2} \log\left( \frac{RL_f L_h}{\epsilon - 2\phi_{Hera}^i L_f L_h} \right) - \frac{\log \delta}{|\mathcal{N}|\epsilon^2} \right). \tag{20}$$

*Here, $\phi_{Hera}^i$ represents the approximation error bound derived from the adaptive rank allocation strategy in FedHera, which satisfies $\phi_{Hera}^i \leq \phi_{Base}^i$ compared to uniform allocation baselines.*

## A.3. Proof of Theorem A.4

*Proof.* The proof adapts the theoretical framework established in FlexLoRA (Bai et al., 2024). We focus on quantifying the impact of the weight approximation error on the global model's generalization performance.

Let $\mathbb{H}^n$ denote the hypothesis space parameterized by the global weights $W_g$. We define the distance metric $\Delta$ to measure the divergence in the expected loss between the original global weights $W_g^i$ and their low-rank approximated counterparts

$W_g^{i,'}$ on client $i$:

$$\Delta(W_g^i, W_g^{i,'}) \triangleq \frac{1}{n} \mathbb{E}_{x,y \sim \mathcal{D}_i} \left[ \left| \sum \left( f(W_g^i; x, y) - \sum f(W_g^{i,'}; x, y) \right) \right| \right]. \tag{21}$$

To bound this distance, we invoke Assumption A.1, the Lipschitz continuity properties. By applying the chain rule and the triangle inequality, we decompose the error induced by the SVD truncation:

$$
\begin{aligned}
\Delta(W_g^i, W_g^{i,'}) &\le L_f \| h_{W_g^i} - h_{W_g^{i,'}} \| \\
&\le L_f L_h \| \text{SVD}(W_g, r^i) - \text{SVD}(W_g', r^i) \| \\
&= L_f L_h \| (\text{SVD}(W_g, r^i) - W_g) + (W_g - W_g') + (W_g' - \text{SVD}(W_g', r^i)) \| \\
&\le L_f L_h \Big( \underbrace{\| \text{SVD}(W_g, r^i) - W_g \|}_{\Phi_1} + \underbrace{\| \text{SVD}(W_g', r^i) - W_g' \|}_{\Phi_2} + \| W_g - W_g' \| \Big).
\end{aligned}
\tag{22}
$$

Applying Assumption A.2, the approximation error terms $\Phi_1$ and $\Phi_2$ are bounded by the constant $\phi^i$. Thus:

$$\Delta(W_g^i, W_g^{i,'}) \le L_f L_h \left[ 2\phi^i + \| W_g - W_g' \| \right]. \tag{23}$$

We now demonstrate that FedHera minimizes the constant $\phi^i$. As elaborated in Section 4.2, FedHera views rank assignment as a global optimization problem. Let $R_{total}$ be the total rank budget. FedHera maximizes the retained energy:

$$
\begin{aligned}
\max_{\{r_1,\ldots,r_L\}} \sum_{l=1}^{L} \sum_{j=1}^{r_l} (\sigma_{l,j})^2 \quad &\text{s.t.} \sum_{l=1}^{L} r_l \le R_{total} \\
\iff \min_{\{r_1,\ldots,r_L\}} \underbrace{\sum_{l=1}^{L} \sum_{j=r_l+1}^{q_l} (\sigma_{l,j})^2}_{\text{Total Residual Energy } \mathcal{E}_{res}} \quad &\text{s.t.} \sum_{l=1}^{L} r_l \le R_{total},
\end{aligned}
\tag{24}
$$

where $q_l = \min(d_l, p_l)$. Based on **Lemma A.3**, minimizing the Frobenius norm approximation error is equivalent to minimizing $\mathcal{E}_{res}$. Since FedHera employs a greedy water-filling strategy that prioritizes the largest singular values globally, it achieves the minimal possible residual energy for the given budget. Therefore, if $\phi_{Hera}$ and $\phi_{Base}$ are the error bounds for FedHera and a baseline, we have:

$$\phi_{Hera} \le \phi_{Base}. \tag{25}$$

Substituting this bound into the sample complexity result from FlexLoRA (Bai et al., 2024), the required sample size is:

$$\tilde{N} = \mathcal{O} \left( \frac{k}{|\mathcal{N}|\epsilon^2} \log \left( \frac{R L_f L_h}{\epsilon - 2\phi^i L_f L_h} \right) - \frac{\log \delta}{|\mathcal{N}|\epsilon^2} \right). \tag{26}$$

The term $\tilde{N}$ is monotonically increasing with respect to $\phi^i$. Specifically, a smaller $\phi^i$ increases the denominator ($\epsilon - 2\phi^i L_f L_h$), thereby reducing the logarithmic term. Since $\phi_{Hera} \le \phi_{Base}$, it follows that $\tilde{N}_{Hera} \le \tilde{N}_{Base}$. This concludes the proof that FedHera requires fewer samples to guarantee the same generalization bound. $\square$

## B. Method Component Comparison

To clarify the methodological positioning of FEDHERA, we compare representative heterogeneous federated LoRA methods along several design dimensions. Here, *LoRA factor-averaging bias* refers to the mismatch caused by directly averaging the low-rank factors before multiplication:

$$\sum_i p_i \mathbf{B}_i \mathbf{A}_i \ne \left( \sum_i p_i \mathbf{B}_i \right) \left( \sum_i p_i \mathbf{A}_i \right). \tag{27}$$

A method is marked as avoiding this bias if it aggregates reconstructed updates, uses stacking-based aggregation, or applies residual/full-rank correction instead of directly averaging $\mathbf{A}$ and $\mathbf{B}$ independently. The last column indicates whether the main mechanism of the method is orthogonal to the choice of server-side aggregation rule.

*Table 3.* **Comparison of heterogeneous federated LoRA methods.** ✓ indicates that the method explicitly supports the corresponding property.

| METHOD | RANK HETEROGENEITY | DECOUPLE $r^{\text{tot}}/r^{\text{train}}$ | AVOIDS LORA FACTOR BIAS | AGGREGATION ORTHOGONAL |
|---|---|---|---|---|
| FLEXLORA (BAI ET AL., 2024) | ✓ | × | ✓ | × |
| FEDHL (PENG ET AL., 2025) | ✓ | × | ✓ | × |
| FLORA (WANG ET AL., 2024) | ✓ | × | ✓ | × |
| HETLORA (CHO ET AL., 2023) | ✓ | × | × | × |
| **FEDHERA** | ✓ | ✓ | ✓ | ✓ |

FlexLoRA, FedHL, FLoRA, and HetLoRA all support heterogeneous client capacities, but they still assign each client a single effective LoRA capacity. Thus, the information a client receives remains tied to what it can locally train and upload.

In contrast, FEDHERA explicitly separates the downloaded rank from the trainable rank. This allows clients to receive a richer global basis while optimizing only a budget-feasible prefix. Moreover, the rank-decoupled client-side mechanism is not tied to a specific server aggregation rule, and can be combined with either standard reconstructed-update aggregation or residual-style aggregation.

## C. Detailed Dataset Statistics and Evaluation Metrics

This section summarizes the datasets, data splits, and evaluation protocols used in our experiments, covering three task categories: Arithmetic Reasoning, Commonsense Reasoning, and Generative tasks. At the beginning of each communication round, we evaluate the aggregated global model on the local evaluation splits of clients not selected for training in that round (held-out, pre-update evaluation). We then average these pre-training metrics across the held-out clients to obtain the round-level performance of the global model. For monitoring purposes, we may also evaluate participating clients before local updates. Unless otherwise stated, all reported results follow the held-out protocol above.

### C.1. Arithmetic Reasoning

We evaluate mathematical reasoning under two benchmarks:

- **GSM8K (Cobbe et al., 2021):** A dataset of grade-school math word problems requiring multi-step reasoning. We fine-tune on the training split and report performance on the standard test split.

- **MetaMathQA (Arithmetic) (Yu et al., 2024):** A large-scale dataset constructed by rewriting math questions from multiple perspectives. We fine-tune on the training set and evaluate on the held-out validation split.

**Task Formulation.** Each example is formatted as an instruction-following instance. Given a problem statement $x$, the model generates a chain-of-thought solution followed by a final answer. We evaluate only the final answer via exact match, intermediate rationales are not scored. To standardize evaluation, we enforce an explicit final-answer marker in the target output. Specifically, the reference output ends with `<final_answer>`, where `<final_answer>` is a canonical numeric string. We also store the canonical final answer in an additional field `final_answer` during preprocessing.

**Exact Match Accuracy.** We compute final-answer exact match accuracy. For each generated response, we extract the final answer and compare it with the gold `final_answer`. A prediction is correct if and only if the normalized extracted answer exactly matches the canonical label.

### C.2. Commonsense Reasoning

We evaluate commonsense reasoning on four benchmarks:

- **BoolQ (Clark et al., 2019):** Yes/no question answering based on a passage.

- **PIQA (Bisk et al., 2020):** Physical commonsense reasoning with two candidate solutions.

- **HellaSwag (Zellers et al., 2019):** Adversarially filtered commonsense NLI with four candidate endings.

- **WinoGrande (Sakaguchi et al., 2021):** Pronoun resolution with a binary choice.

**Task Formulation.** We cast all benchmarks as label prediction via instruction following. Each example provides the context and candidate options, and the model is instructed to output only the option label. Concretely, the target outputs are standardized as: `True`/`False` for BoolQ, `A`/`B` for PIQA and WinoGrande, and `A`/`B`/`C`/`D` for HellaSwag.

**Accuracy.** We compute task accuracy from greedy generations. The model prediction is the first valid label token extracted from the generated text. We normalize outputs by stripping whitespace and punctuation before extracting the first valid label. Accuracy is the proportion of examples where the predicted label matches the gold label, averaged over each client's local evaluation set and aggregated across clients.

### C.3. Generative Tasks

We evaluate generation capabilities on two benchmarks:

- **E2E NLG (Novikova et al., 2017):** A data-to-text generation task where the model produces fluent restaurant descriptions from meaning representations.

- **Alpaca (Taori et al., 2023):** An instruction-following dataset covering diverse tasks such as summarization, creative writing, and information extraction.

**Task Formulation.** We formulate both benchmarks as sequence-to-sequence generation tasks. For E2E NLG, the input is a structured meaning representation, and the model is instructed to generate a corresponding natural language description. For Alpaca, the input consists of an instruction, and the model generates a direct response. In both cases, we use standard causal language modeling objectives during training and greedy decoding during evaluation.

**Metrics.** For E2E NLG, we report evaluation loss, BLEU, ROUGE-L, METEOR, NIST, and CIDEr scores compared to human references. For Alpaca, we evaluate the generated responses using evaluation loss, ROUGE-L, and BLEU against the reference outputs to measure fluency and content overlap.

Unless otherwise stated, we simulate $N = 100$ clients by randomly shuffling each dataset and uniformly partitioning the resulting examples across clients, yielding an approximately IID data distribution across clients. Each client then splits its local shard into train/eval/test with a ratio of 0.8/0.1/0.1. This controlled partitioning isolates the effect of Uniform or Skewed resource heterogeneity from data heterogeneity. With uniform partitioning, each client receives either $\lfloor |\mathcal{D}|/N \rfloor$ or $\lceil |\mathcal{D}|/N \rceil$ examples for a dataset of size $|\mathcal{D}|$. For large datasets, e.g., MetaMathQA, we optionally subsample up to $M$ examples before partitioning. $M$ is specified by `--max_examples`.

## D. Resource Heterogeneity and Baselines.

### D.1. Resource Profiles

We model system heterogeneity for each client $i$ using a resource constraint triplet $[B_i, M_i, T_i]$, representing downlink bandwidth, available VRAM, and (optionally) a local training latency budget. Unless otherwise stated, our experiments primarily vary $B_i$ and $M_i$ to study the communication–optimization asymmetry under heterogeneous resources. We evaluate under two resource-heterogeneity profiles:

- **Uniform Distribution:** Client resource budgets have low variance, and bandwidth, memory, and compute capabilities are of comparable magnitude across devices. This setting serves as a stability check in low-heterogeneity environments.

- **Skewed Distribution:** Client budgets are highly imbalanced, with a majority of mid-tier devices and a smaller fraction of low-end and high-end outliers. This profile stress-tests whether an algorithm can effectively decouple communication and training constraints to improve resource utilization.

We use a controlled, tiered heterogeneity model with three resource levels (low/medium/high). Rather than relying on hardware-specific bandwidth/latency measurements, we calibrate rank-equivalent budgets so that each tier corresponds to a

*Table 4.* Tiered resource profiles used in our simulator. Budgets are calibrated to match canonical rank capabilities.

| Tier | $r^{\text{train}}$ (train) | $r^{\text{tot}}$ (download) |
|---|---|---|
| Low | 4 | 32 |
| Medium | 8 | 48 |
| High | 16 | 64 |

| Profile | $p(\text{low})$ | $p(\text{med})$ | $p(\text{high})$ |
|---|---|---|---|
| Uniform | 1/3 | 1/3 | 1/3 |
| Skewed | 0.3 | 0.5 | 0.2 |

canonical LoRA capability. Concretely, we set target ranks $(r^{\text{train}}, r^{\text{tot}}) \in \{(4, 32), (8, 48), (16, 64)\}$ for (low/medium/high) tiers, and derive the effective download budget $B_i$ and step-time budget $T_i$ by scaling with the per-rank parameter footprint estimated from the active LoRA layers. This normalization makes the resource profiles comparable across different backbone models and target-module choices. Uniform uses a balanced tier mixture with probabilities $(1/3, 1/3, 1/3)$, whereas Skewed uses an imbalanced mixture $(0.3, 0.5, 0.2)$ dominated by the medium tier. We treat VRAM as a normalization constant in our simulator and focus on heterogeneity in effective training capacity (captured by $r^{\text{train}}$ and the step-time budget) versus download capacity (captured by $r^{\text{tot}}$ and $B_i$), which is the main asymmetry studied in this work.

### D.2. Baseline Descriptions

We benchmark FEDHERA against a homogeneous baseline and SOTA heterogeneous federated fine-tuning approaches:

- **Homogeneous-rank baseline (FedHomoLoRA):** We denote this conventional baseline as **FedHomoLoRA** for clarity. This approach represents a homogeneous baseline adapted for heterogeneous settings. To accommodate the client with the most limited resources, all participants are forced to adhere to a fixed and uniform rank configuration. The server aggregates the global model by performing component-wise averaging of the low-rank matrices $A$ and $B$. This method highlights the performance trade-off incurred when reducing the system capacity to the lowest common denominator.

- **FlexLoRA (Bai et al., 2024):** This is a heterogeneous LoRA method that aggregates the reconstructed weight updates $\Delta W$ and employs SVD on the global update to redistribute adapters. Although FlexLoRA supports variable ranks across clients, it employs a coupled resource strategy where the local training rank is strictly tied to the communication rank. This coupling limits flexibility when bandwidth and computation constraints diverge.

- **FedHL (Peng et al., 2025):** FedHL mitigates truncation-induced bias in heterogeneous low-rank aggregation by using the current global model as an aggregation baseline and aggregating low-rank residual updates, thereby reducing the bias introduced by client-specific truncation. It further reweights client contributions based on their estimated truncation/approximation error to down-weight drifted updates.

- **FedHeLLo (Zhang et al., 2025):** This framework addresses resource heterogeneity through layer-wise pruning rather than rank adaptation. Clients train only a specific subset of model layers determined by their computational capacity, while remaining layers stay frozen. The server aggregates updates by averaging the overlapping parameters from different clients. This baseline contrasts rank-based heterogeneity with depth-based or width-based heterogeneity solutions.

## E. Implementation Details

### E.1. Federated Simulation Setup

We simulate federated fine-tuning with heterogeneous client resource budgets. Unless otherwise stated, we use $N = 100$ clients. In each communication round, we randomly sample a fraction $C = 0.1$ of clients to participate in local training, following the standard partial-participation protocol in FL. We run $R = 10$ communication rounds. All results are averaged over three runs with different random seeds.

### E.2. Hyperparameters and Training Configuration

All models are trained using the AdamW optimizer with a cosine learning-rate schedule (no warm-up).

**Task-Specific Evaluation Settings.** For reasoning tasks, we evaluate final-answer exact match accuracy using answer extraction (Appendix C). When reporting evaluation loss, we optionally compute it on the answer span only (enabled by `--eval_answer_only_loss`) to obtain a token-level likelihood metric focused on the predicted answer, which complements exact match. For generative tasks, we compute loss on the full sequence.

**LoRA Configuration.** For coupled baselines and the homogeneous-rank baseline, we use a default LoRA configuration with rank $r = 8$, scaling factor $\alpha = 16$, and dropout $p = 0.05$, unless otherwise stated. The default target modules include the attention projections (e.g., $W_q, W_k, W_v$ / q_proj, k_proj, v_proj); additional target-module variants are evaluated in Appendix F.9.

Table 5 details the hyperparameter settings for Mathematical Reasoning (Mistral-7B), Commonsense Reasoning (LLaMA-2-7B), and Generative Tasks (Llama-3.2-3B). Table 6 reports the configuration for Qwen3-0.6B used in the drift analysis (Section 6).

*Table 5.* Hyperparameter settings for the main evaluation benchmarks: Mathematical Reasoning (Mistral-7B), Commonsense Reasoning (Llama-2-7B), and Generative Tasks (Llama-3.2-3B).

| Task Category | Math Reasoning | Commonsense Reasoning | Generative Task |
|---|---|---|---|
| **Base Model** | Mistral-7B-Instruct-v0.2 | Llama-2-7b-hf | Llama-3.2-3B-Instruct |
| **Optimizer** | AdamW | AdamW | AdamW |
| **Local Learning Rate** | $2 \times 10^{-5}$ | $2 \times 10^{-5}$ | $2 \times 10^{-5}$ |
| **LR Scheduler** | Cosine | Cosine | Cosine |
| **Local Batch Size** | 4 | 4 | 4 |
| **Local Epochs** | 5 | 5 | 5 |
| **Max Seq Length** | 512 | 512 | 512 |
| **LoRA Rank** ($r$) | 8 | 8 | 8 |
| **LoRA Alpha** ($\alpha$) | 16 | 16 | 16 |
| **LoRA Dropout** | 0.05 | 0.05 | 0.05 |
| **Target Modules** | [q_proj, k_proj, v_proj] | [q_proj, k_proj, v_proj] | [q_proj, k_proj, v_proj] |

*Table 6.* Hyperparameter settings for the Drift Dynamics Analysis using Qwen3-0.6B. A larger batch size and learning rate were employed to rigorously test optimization stability.

| Configuration | Drift Analysis Setting |
|---|---|
| **Base Model** | Qwen/Qwen3-0.6B |
| **Optimizer** | AdamW |
| **Local Learning Rate** | $5 \times 10^{-5}$ |
| **LR Scheduler** | Cosine |
| **Local Batch Size** | 16 |
| **Micro Batch Size** | 8 |
| **Local Epochs** | 5 |
| **Max Seq Length** | 512 |
| **LoRA Rank** ($r$) | 8 |
| **LoRA Alpha** ($\alpha$) | 16 |
| **Target Modules** | Default (Attn) |

# F. Additional Experimental Results

## F.1. Additional Results on Reasoning Tasks (Uniform Distribution)

In Section 5.2, we focused on the Skewed heterogeneity profile to highlight the resilience of FEDHERA under extreme resource disparities. Here, we provide the corresponding results for the Uniform distribution, where client resources are distributed with lower variance.

**Results.** Figure 6 illustrates the accuracy gain over the homogeneous baseline across the six reasoning benchmarks under the Uniform setting. Although the resource gap between clients is narrower in this scenario, FEDHERA still consistently outperforms the heterogeneous baselines. This demonstrates that our energy-aware water-filling strategy is not merely a mitigation for extreme heterogeneity; rather, it provides a more fundamentally efficient mechanism for utilizing available bandwidth. Even when clients have moderate capacities, decoupling the downloaded rank from the training rank allows for a more expressive global basis retention, leading to better global-model generalization.

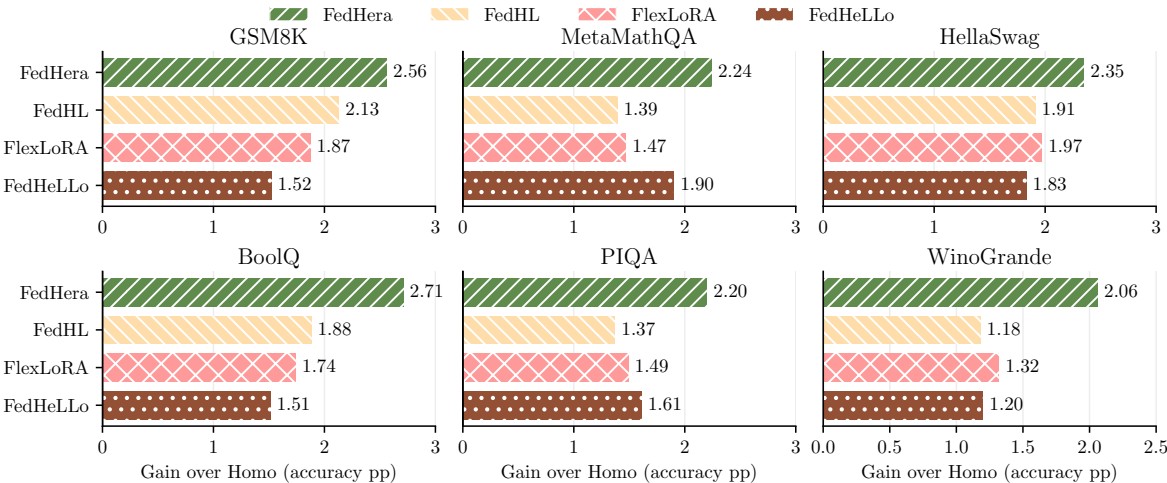

*Figure 6.* **Performance gain on reasoning benchmarks (Uniform distribution).** Accuracy improvement (percentage points) over the homogeneous baseline across six reasoning tasks.

## F.2. Final-Round Global Model Performance

In this section, we report the absolute performance metrics of the final-round global model across all tasks and heterogeneity profiles. Final-round results are evaluated on all clients' local test splits and averaged across clients. While the main text reports gains over the homogeneous-rank baseline to normalize for task difficulty, the tables below provide the raw values for direct comparison.

**Reasoning Tasks (Accuracy).** Table 7 summarizes the final test accuracy (%) for the arithmetic and commonsense reasoning benchmarks. FEDHERA achieves the best accuracy in the majority of settings, confirming its robustness across tasks and heterogeneity profiles.

**Generative Tasks (Comprehensive Metrics).** Tables 8 and 9 present the comprehensive evaluation results for the generative tasks under *Skewed* and *Uniform* distributions, respectively. Unlike the main text which focuses on ROUGE-L for brevity, here we report a broader suite of metrics to strictly assess generation quality:

- **Alpaca (Instruction Tuning):** We report ROUGE-L and BLEU to measure the overlap with reference responses.

- **E2E NLG (Data-to-Text):** In addition to ROUGE-L and BLEU, we include semantic metrics including METEOR, NIST, and CIDEr. These metrics are crucial for evaluating data-to-text fidelity, ensuring that the generated descriptions accurately cover the input meaning representations without hallucination or omission.

*Table 7.* Final-round test accuracy (%) on Reasoning benchmarks under Uniform and Skewed distributions. **Bold** indicates the best performance.

| METHOD | GSM8K UNIF. | GSM8K SKEW. | METAMATH UNIF. | METAMATH SKEW. | HELLASWAG UNIF. | HELLASWAG SKEW. | BOOLQ UNIF. | BOOLQ SKEW. | PIQA UNIF. | PIQA SKEW. | WINOGRANDE UNIF. | WINOGRANDE SKEW. |
|---|---|---|---|---|---|---|---|---|---|---|---|---|
| FEDHOMOLORA | 46.4 | 45.6 | 45.0 | 44.3 | 56.1 | 55.5 | 39.9 | 39.3 | 56.5 | 56.1 | 48.7 | 48.1 |
| FLEXLORA | 45.8 | 45.7 | 45.2 | 45.0 | 56.5 | 56.3 | 41.5 | 41.1 | 57.2 | 56.9 | 49.2 | 48.8 |
| FEDHL | 46.9 | 46.3 | 45.7 | 45.3 | 57.8 | 57.4 | 41.6 | 41.4 | 57.3 | 57.3 | 49.3 | 49.2 |
| FEDHELLO | 46.7 | 46.2 | 45.5 | 44.9 | 57.9 | 57.4 | 41.9 | 41.6 | 57.5 | 57.4 | 49.3 | 49.1 |
| **FEDHERA** | **47.6** | **47.1** | **46.4** | **46.2** | **58.8** | **58.4** | **42.2** | **42.2** | **58.2** | **58.0** | **51.8** | **51.7** |

Consistent with the reasoning results, FEDHERA achieves superior scores across all semantic metrics, particularly in the challenging Skewed setting (Table 8), confirming that our decoupled strategy better preserves the semantic integrity of the global model.

*Table 8.* **Generative Task Results (Skewed distribution).** Performance comparison on Alpaca and E2E NLG under the skewed resource distribution.

| METHOD | ALPACA ROUGEL | ALPACA BLEU | E2E NLG ROUGEL | E2E NLG BLEU | E2E NLG MET. | E2E NLG NIST | E2E NLG CIDER |
|---|---|---|---|---|---|---|---|
| FEDHOMOLORA | 0.608 | 0.173 | 0.680 | 0.425 | 0.670 | 4.336 | 0.114 |
| FLEXLORA | 0.698 | 0.224 | 0.862 | 0.641 | 0.870 | 5.696 | 0.340 |
| FEDHL | 0.699 | 0.226 | 0.865 | 0.645 | 0.871 | 5.779 | 0.347 |
| FEDHELLO | 0.611 | 0.181 | 0.730 | 0.450 | 0.712 | 4.618 | 0.121 |
| **FEDHERA** | **0.709** | **0.231** | **0.875** | **0.650** | **0.882** | **5.883** | **0.440** |

*Table 9.* **Generative Task Results (Uniform distribution).** Performance comparison on Alpaca and E2E NLG under the Uniform setting.

| METHOD | ALPACA ROUGEL | ALPACA BLEU | E2E NLG ROUGEL | E2E NLG BLEU | E2E NLG MET. | E2E NLG NIST | E2E NLG CIDER |
|---|---|---|---|---|---|---|---|
| FEDHOMOLORA | 0.609 | 0.175 | 0.684 | 0.430 | 0.677 | 4.490 | 0.176 |
| FLEXLORA | 0.701 | 0.227 | 0.864 | 0.642 | 0.872 | 5.722 | 0.345 |
| FEDHL | 0.703 | 0.230 | 0.871 | 0.649 | 0.875 | 5.801 | 0.350 |
| FEDHELLO | 0.620 | 0.195 | 0.741 | 0.463 | 0.723 | 4.776 | 0.202 |
| **FEDHERA** | **0.708** | **0.236** | **0.879** | **0.652** | **0.883** | **5.905** | **0.447** |

## F.3. Robustness to Non-IID Data Distributions

While the main experiments utilize near-IID partitions to isolate system heterogeneity, real-world federated deployments often face statistical heterogeneity. To verify the robustness of FEDHERA under such conditions, we performed a stress test on the Alpaca instruction-tuning task. We construct non-IID partitions by first clustering instructions via $K$-means based on semantic embeddings and then sampling clusters for each client using a Dirichlet distribution with concentration parameter $\alpha \in \{10, 0.5, 0.1\}$. A smaller $\alpha$ implies that a client's local data is dominated by fewer instruction clusters, simulating severe task skew.

We retain the Skewed resource distribution and follow the held-out evaluation protocol (Section 5.2). Figure 7 visualizes the performance trajectory, and Table 10 summarizes the final-round ROUGE-L scores.

**Results and Discussion.** By increasing data heterogeneity generally degrades performance across all methods due to the divergence of local optimization directions. However, FEDHERA *consistently* outperforms the baselines in all settings. Notably, as shown in Table 10, FEDHERA maintains a solid performance advantage even under the high-skew data setting ($\alpha = 0.1$).

We attribute this robustness to the decoupled dual-rank design. In coupled methods, e.g., FlexLoRA, severe data skew

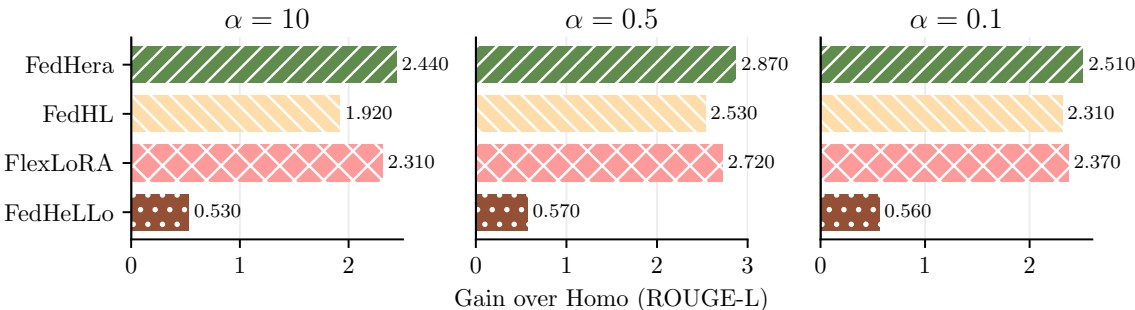

*Figure 7.* **Robustness under Data Heterogeneity.** ROUGE-L improvement of heterogeneous methods over the homogeneous baseline across varying Dirichlet concentration parameters $\alpha \in \{10, 0.5, 0.1\}$ on the Alpaca dataset. The resource distribution is fixed to Skewed.

*Table 10.* **Final-round ROUGE-L under Non-IID Alpaca.** Performance comparison across varying degrees of data heterogeneity ($\alpha$).

| METHOD | $\alpha = 10$ | $\alpha = 0.5$ | $\alpha = 0.1$ |
|---|---|---|---|
| FLEXLORA | 0.468 | 0.463 | 0.460 |
| FEDHL | 0.460 | 0.455 | 0.454 |
| FEDHELLO | 0.456 | 0.459 | 0.445 |
| **FEDHERA** | **0.479** | **0.475** | **0.471** |

causes the locally trained low-rank adapters to overfit to local tasks, leading to aggregation conflict. In contrast, FEDHERA allows clients to download a higher-rank global basis ($r^{\text{tot}}$) that acts as a stable spectral anchor during the forward pass. This frozen global context effectively regularizes local training, preventing the trainable prefix from drifting too far towards client-specific distributions.

This ablation serves as a preliminary verification of robustness. While FEDHERA demonstrates resilience to statistical skew, tackling extreme non-IID distributions with specialized data-efficient tuning techniques (e.g., (Qin et al., 2025)) represents a complementary direction for future work.

### F.4. Additional Federation-Scale Diagnostics

To complement the main evaluation, we conduct an additional diagnostic study on larger federation scales. The goal is not to provide an exhaustive scaling benchmark, but to examine whether the drift-related advantage of FEDHERA persists when the federation becomes larger and more fragmented. We use Qwen3-0.6B on E2E NLG and run each method for 20 communication rounds. We vary the total number of clients $N \in \{100, 300, 500\}$, while keeping the number of participating clients per round fixed. This setting isolates the effect of increasing the federation size, rather than simply increasing the number of aggregated client updates per round. We also include FLoRA (Wang et al., 2024), a related rank-heterogeneous LoRA baseline, as an additional rank-heterogeneous LoRA baseline in this comparison. For each setting, we report evaluation loss, ROUGE-L, cross-client drift, and absolute drift.

Table 11 shows that all methods become more challenging to optimize as $N$ increases and the participating clients represent a smaller fraction of the full federation. Nevertheless, FEDHERA remains consistently competitive across all tracked quantities and achieves the best result in each scale setting. FEDHERA also maintains lower cross-client drift and lower absolute drift, suggesting that the decoupled rank design continues to provide a useful shared spectral reference as the federation becomes more fragmented.

### F.5. Impact of Water-Filling Rank Allocation

A pivotal component of FEDHERA's server-side design is its spectral-based rank allocation. Rather than assigning a uniform rank per layer, our water-filling strategy formulates rank assignment as a global budgeting problem: it pools singular values across layers and allocates rank to directions with higher spectral energy, regardless of layer index. To validate the necessity of this design, we compare against two allocation variants under matched communication and training budgets:

- **Uniform Allocation:** For each client $i$, we choose the largest single rank $r$ that satisfies the client's budgets when

*Table 11.* **Additional federation-scale diagnostics on Qwen3-0.6B / E2E NLG.** The number of participating clients per round is fixed while the total number of clients increases.

| $N$ | METHOD | LOSS ↓ | ROUGE-L ↑ | CROSS DRIFT ↓ | ABS. DRIFT ↓ |
|---|---|---|---|---|---|
| | **FEDHERA** | **0.5787** | **0.7899** | **0.4855** | **2.0601** |
| | FEDHL | 0.6016 | 0.7731 | 0.5493 | 2.0724 |
| 100 | FLEXLORA | 0.6581 | 0.7745 | 0.6864 | 2.9912 |
| | FLORA | 0.6416 | 0.7726 | 0.6398 | 2.4160 |
| | FEDHELLO | 0.7383 | 0.7663 | 0.6796 | 4.1703 |
| | **FEDHERA** | **0.6075** | **0.7822** | **0.5780** | **2.1138** |
| | FEDHL | 0.6350 | 0.7623 | 0.5829 | 2.1315 |
| 300 | FLEXLORA | 0.6610 | 0.7616 | 0.7092 | 3.1040 |
| | FLORA | 0.6663 | 0.7656 | 0.6526 | 2.7088 |
| | FEDHELLO | 0.7464 | 0.7578 | 0.7221 | 4.1992 |
| | **FEDHERA** | **0.6152** | **0.7710** | **0.6279** | **2.1455** |
| | FEDHL | 0.6551 | 0.7615 | 0.6299 | 2.1868 |
| 500 | FLEXLORA | 0.7145 | 0.7585 | 0.7509 | 3.1588 |
| | FLORA | 0.6838 | 0.7649 | 0.6945 | 2.8146 |
| | FEDHELLO | 0.7492 | 0.7572 | 0.7896 | 4.2087 |

applied identically to all target layers. This coarse-grained baseline mirrors the common practice of using a uniform rank across layers and ignores layer-wise differences in spectral sensitivity.

- **Random Allocation:** We randomly assign layer-wise ranks $\{r_{\ell,i}\}$ while keeping the total budget usage matched to the client, both communication volume and trainable-parameter footprint. This control decouples the effect of parameter quantity from allocation quality.

**Results.** We conduct this ablation on LLaMA-2-7B for WinoGrande. As shown in Table 12, water-filling achieves the best performance. Uniform allocation underperforms (49.8%), suggesting that a single-rank choice cannot simultaneously fit layers with different spectral decay. Random allocation further degrades accuracy (49.0%), indicating that merely matching the budget without prioritizing informative directions is ineffective. In contrast, FEDHERA reaches **51.8%**, demonstrating that global competition for rank resources improves budget efficiency by preserving high-energy spectral components.

*Table 12.* **Ablation Study of Rank Allocation Strategies.** Comparison of test accuracy and evaluation loss on the **WinoGrande** dataset. FEDHERA's global energy-aware allocation significantly outperforms uniform and random baselines.

| ALLOCATION STRATEGY | ACCURACY ↑ | LOSS ↓ |
|---|---|---|
| RANDOM ALLOCATION | 49.0% | 0.365 |
| UNIFORM ALLOCATION | 49.8% | 0.359 |
| **FEDHERA** | **51.8%** | **0.346** |

### F.6. Disentangling the Impact of Unbiased Aggregation

While FEDHERA primarily targets client-side optimization stability via dual-rank decoupling and adaptive warm-up, the server-side aggregation strategy plays a supportive role in reconciling heterogeneous updates. To explicitly attribute the performance gains and verify the robustness of our framework, we isolate the impact of the residual-based aggregation strategy (Peng et al., 2025) by comparing it against a standard baseline:

- **Standard Aggregation:** The server aggregates the reconstructed updates directly: $\mathbf{W}_g^{t+1} = \mathbf{W}_g^t + \sum p_i \mathbf{B}_i^{\mathrm{up}} \mathbf{A}_i^{\mathrm{up}}$. This approach aligns with traditional FedAvg-style aggregation but introduces systematic bias due to the loss of information in the null space of the truncated adapters.

- **Unbiased Aggregation (Default):** The strategy employed in FEDHERA (Eq. 10), which explicitly subtracts the initialization terms to strictly preserve the components not updated by clients, thereby ensuring an unbiased optimization trajectory.

**Results.** We perform this ablation on LLaMA-2-7B for WinoGrande, maintaining the identical resource constraints and hyperparameters as in the main experiments. As detailed in Table 13, the residual-based aggregation yields a performance boost of +0.8% in accuracy. Crucially, even with Standard Aggregation, FEDHERA achieves a competitive accuracy of 51.0%, which significantly outperforms the heterogeneous baselines reported in Section 5.2. This confirms that while unbiased aggregation serves as a valuable stabilization plug-in, the primary performance drivers of FEDHERA are the resource-aware rank allocation and the prefix-gated training mechanism.

| Aggregation Strategy | Accuracy ↑ | Loss ↓ |
|---|---|---|
| FEDHERA w/ Standard Agg. | 51.0% | 0.351 |
| **FEDHERA (Default Unbiased)** | **51.8%** | **0.346** |

*Table 13.* Ablation study on server-side aggregation strategies.

### F.7. Fine-grained Resource Utilization and System Efficiency

To verify that FEDHERA's gains come from improved resource utilization rather than simply increasing trainable capacity, we analyze efficiency from two perspectives: (1) **Memory Saturation**, which measures how effectively the method utilizes a bounded training-memory budget, and (2) **Information Throughput**, which quantifies how much adapter capacity participates in the forward pass per unit of trainable footprint.

**Experimental Setup.** We evaluate system efficiency on GSM8K using Mistral-7B under a simulated resource-constrained setting. For each client, we enforce a fixed training-memory budget and record the peak GPU memory allocated during local training. We compare against strong heterogeneous baselines.

**1. Granularity and Memory Saturation.** A limitation of coarse-grained baselines is budget fragmentation: using a single uniform rank across layers can leave residual memory that is insufficient for increasing the global rank, resulting in under-utilization. In contrast, FEDHERA's layer-wise water-filling allocates trainable ranks $\{r_{i,\ell}^{train}\}$ at finer granularity, reducing fragmentation and more tightly saturating the budget.

**Metric.** We report the VRAM Utilization Rate as $(M_{\text{peak}}/M_i) \times 100\%$, where values closer to 100% indicate better saturation of the training-memory budget.

**Results.** As shown in Figure 8(a), FedHeLLo and FlexLoRA reach 90.9% and 93.2% average utilization, respectively, leaving noticeable gaps. FEDHERA improves utilization to 98.7%, indicating substantially reduced fragmentation under the same budget.

**2. Decoupling and Information Throughput.** Coupled baselines enforce $r^{\text{train}} \equiv r^{\text{tot}}$, so the forward-pass adapter capacity is strictly bottlenecked by the expensive optimizer-state footprint. FEDHERA decouples these constraints by downloading a higher-rank basis while training only a budget-feasible prefix ($r^{\text{tot}} > r^{\text{train}}$).

**Metric.** We report the Information Gain Ratio (IGR) as the ratio of adapter parameters participating in the forward pass (determined by $r^{\text{tot}}$) to the adapter parameters that are trainable and require optimizer states (determined by $r^{\text{train}}$).

**Results.** While coupled baselines normalize to 1.0, FEDHERA achieves an average IGR of 2.27× (Figure 8(b)). This shows that, per unit of training-memory footprint, FEDHERA leverages substantially more downloaded basis capacity during forward computation, which helps stabilize optimization and mitigate truncation-induced drift.

### F.8. Impact of Adaptive Tail Warm-up (ATW)

The ATW mechanism is motivated by the observation that the server-side SVD-derived global basis stabilizes progressively across rounds. In early communication rounds, the aggregated basis is estimated from high-variance heterogeneous updates, and the frozen tail may contain noise or client-specific artifacts rather than transferable directions. If this immature tail is injected into the forward pass with full strength, it can distort gradient signals and increase the deviation of local updates from a high-rank reference trajectory.

**Experimental Setup.** We evaluate this effect via a drift analysis using **Qwen3-0.6B** on **E2E NLG**. For each client, we compute the absolute drift $\delta^{abs} = \|\Delta W_i - \Delta W_i^{\star}\|_F$, where $\Delta W_i^{\star}$ is a high-rank reference update obtained with $r = 1024$. We compare two variants over the first 10 communication rounds:

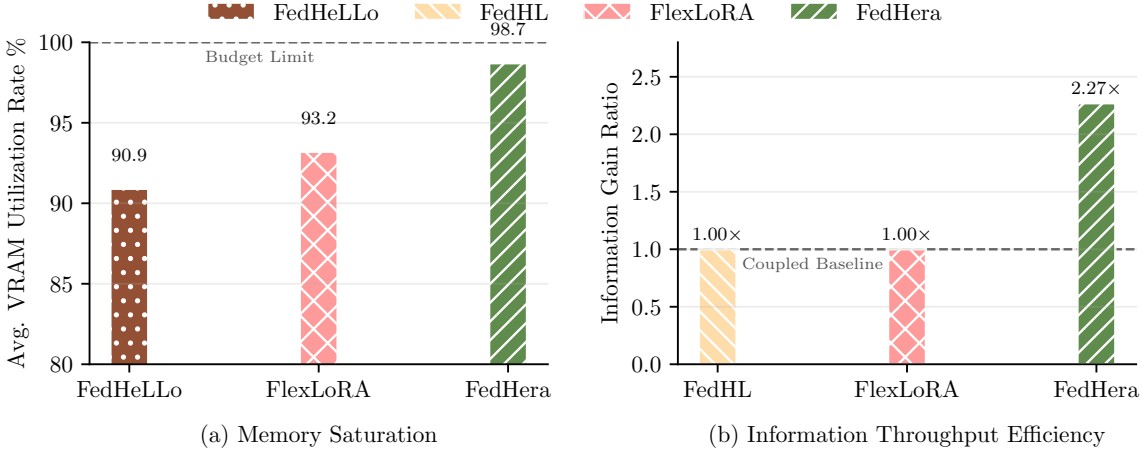

(a) Memory Saturation            (b) Information Throughput Efficiency

*Figure 8.* **System Efficiency Analysis.** Comparison between FedHera and coupled heterogeneous baselines: (a) VRAM Utilization Rate; (b) Information Gain Ratio (IGR), evaluated on GSM8K with Mistral-7B.

- **FedHera (Static tail, $\lambda = 1$):** the frozen tail is fully active from the start;

- **FedHera (ATW):** the tail contribution is adaptively gated by $\lambda$.

**Results and Analysis.** Figure 9 shows that the static-tail variant exhibits substantially larger drift in early rounds, indicating that an immature high-rank tail can amplify projection mismatch and deviate local updates from the reference direction. In contrast, ATW maintains lower drift during this volatile phase by suppressing the tail when alignment is weak, restricting optimization to a lower-rank but higher-fidelity subspace. As the global basis stabilizes, ATW gradually increases tail influence, recovering additional capacity without destabilizing early-round optimization.

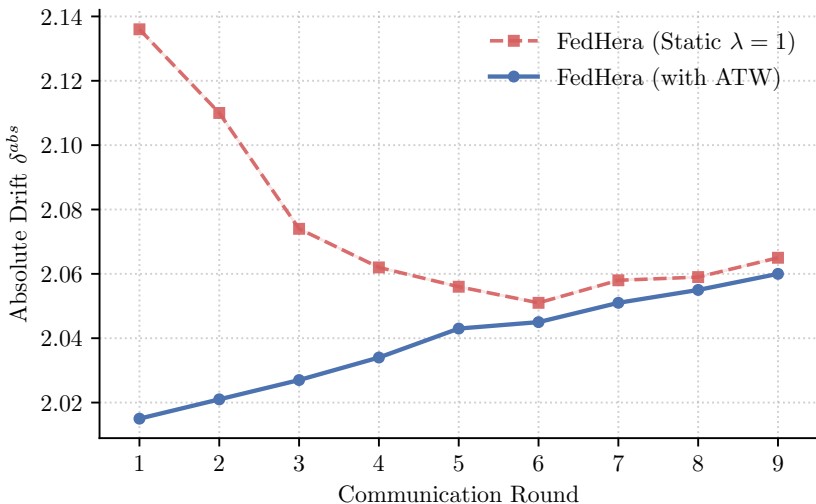

*Figure 9.* **Drift Mitigation via Adaptive Tail Warm-up.** Comparison of absolute drift ($\delta^{abs}$) trajectories on the E2E NLG task (Qwen3-0.6B) over the initial communication rounds.

### F.9. Impact of LoRA Target Modules across Model Scales

To determine the optimal search space for our rank allocation algorithm, we investigate the impact of applying LoRA to different subsets of model parameters. We recognize that the optimal configuration may vary with model capacity. Therefore, we conduct this ablation on two distinct architectures: the lightweight **Qwen3-0.6B** and the larger **Mistral-7B**, measuring the

final evaluation loss on MetaMathQA. We use evaluation loss here as a token-level likelihood metric to compare adaptation scope under matched training budgets. We compare two target configurations:

- **Attention-Only:** Adapters are applied exclusively to the self-attention projections ($\mathbf{W}_q, \mathbf{W}_k, \mathbf{W}_v, \mathbf{W}_o$).

- **All Modules:** Adapters are applied to all available linear layers within the transformer blocks, encompassing both the Attention mechanism and the Feed-Forward Network layers ($\mathbf{W}_{gate}, \mathbf{W}_{up}, \mathbf{W}_{down}$).

*Table 14.* **Ablation on LoRA Target Modules across Model Scales.** Final evaluation loss on **MetaMathQA**. Expanding to All Modules yields larger gains for the smaller model (Qwen3-0.6B) compared to the larger model (Mistral-7B).

| MODEL | TARGET MODULES | FINAL LOSS ↓ |
|---|---|---|
| **QWEN3-0.6B** | ATTN-ONLY | 0.622 |
| | **ALL MODULES** | **0.582** |
| **MISTRAL-7B** | ATTN-ONLY | 0.574 |
| | **ALL MODULES** | **0.566** |

**Results and Analysis.** As shown in Table 14, expanding the adaptation scope to **All Modules** consistently reduces evaluation loss, though the magnitude of improvement varies by model scale. For the smaller Qwen3-0.6B, unlocking the FFN layers yields a substantial performance boost, suggesting that lightweight models rely heavily on the full parameter space to overcome representation bottlenecks in complex reasoning tasks. Conversely, for the larger Mistral-7B, the improvement is positive but more marginal, indicating a phenomenon of diminishing returns where the robust attention mechanism alone captures the majority of the necessary task adaptation.

Although adapting all linear layers yields the best loss, it also increases the optimizer-state footprint substantially. To keep training budgets comparable across methods in the main experiments and to follow common LoRA practice, we adopt the **Attention-Only** configuration by default; the All-Modules setting is reported here as an ablation.

## F.10. Auxiliary Reference Points under Skewed Resources

We additionally report two auxiliary reference settings on the main generative backbone, Llama-3.2-3B, under the skewed resource profile to contextualize the operating range of FEDHERA. The abstention setting represents an extreme resource case where the most constrained clients have $r^{\text{train}} = 0$ and therefore do not contribute local updates. The high-rank oracle-like setting relaxes the training-rank constraint and serves as a ceiling reference rather than a deployable resource-constrained baseline. These references are not used as primary baselines, but help illustrate the gap between a conservative practical alternative and a much less constrained high-rank reference.

*Table 15.* **Auxiliary reference points on Llama-3.2-3B under the skewed resource setting.** Each entry reports loss / ROUGE-L.

| SETTING | ALPACA | E2E NLG |
|---|---|---|
| ABSTENTION REFERENCE | 1.881 / 0.603 | 2.026 / 0.672 |
| FEDHERA | **1.140 / 0.709** | **0.493 / 0.875** |
| HIGH-RANK ORACLE-LIKE REFERENCE | 1.089 / 0.721 | 0.459 / 0.894 |

## F.11. Cross-Client Drift Analysis

Section 6.1 measures local projection drift to show how far a federated low-rank update deviates from a high-rank local reference. Here we examine a complementary effect: cross-client drift, which measures whether heterogeneous clients remain directionally aligned with one another after local training. This is important under resource heterogeneity, since clients may receive different effective global subspaces and optimize under different trainable-rank budgets.

**Metric.** Directly comparing LoRA factors is not meaningful when clients use different ranks. We therefore reconstruct each participating client's residual update in a common dense parameter space. For client $i$, layer $\ell$, and round $t$, we define

$$\mathbf{R}_{i,\ell}^{(t)} = s_{i,\ell}^{\text{up}} \mathbf{B}_{i,\ell}^{\text{up}} \mathbf{A}_{i,\ell}^{\text{up}} - s_{i,\ell}^{\text{init}} \mathbf{B}_{i,\ell}^{\text{init}} \mathbf{A}_{i,\ell}^{\text{init}}, \tag{28}$$

where $s_{i,\ell} = \alpha/r_{i,\ell}$ is the LoRA scaling factor. We concatenate all reconstructed layer-wise residuals into a single dense update vector $\mathbf{r}_i^{(t)} = \text{vec}(\{\mathbf{R}_{i,\ell}^{(t)}\}_\ell)$. The round-wise cross-client drift is then computed as the average pairwise directional dispersion:

$$D_{\text{cross}}^{(t)} = \frac{2}{|\mathcal{S}^{(t)}|(|\mathcal{S}^{(t)}| - 1)} \sum_{i<j,\, i,j \in \mathcal{S}^{(t)}} \left( 1 - \frac{\langle \mathbf{r}_i^{(t)}, \mathbf{r}_j^{(t)} \rangle}{\|\mathbf{r}_i^{(t)}\|_2 \|\mathbf{r}_j^{(t)}\|_2 + \epsilon} \right). \tag{29}$$

Lower values indicate stronger directional agreement among heterogeneous client updates.

**Results.** Figure 10 reports the evolution of $D_{\text{cross}}^{(t)}$ over communication rounds. FEDHERA consistently yields lower cross-client drift than FEDHL and FLEXLORA. This suggests that decoupling the received rank from the trainable rank does not introduce additional inter-client disagreement. Instead, the richer received global basis provides a shared spectral reference that helps align local update directions across heterogeneous clients.

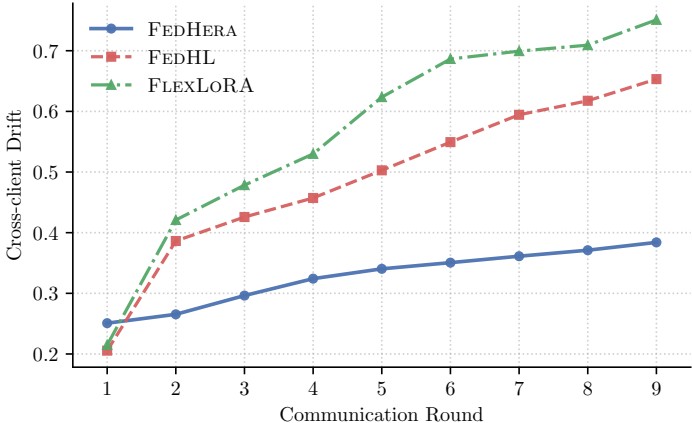

*Figure 10.* **Cross-client drift over communication rounds.** Directional dispersion between reconstructed client residual updates; lower values indicate stronger cross-client alignment.

These results complement the oracle-based drift analysis in Section 6.1. While Section 6.1 shows that FEDHERA reduces deviation from a high-rank local reference, Figure 10 further shows that it improves directional consistency across clients. Coupled heterogeneous baselines restrict each client to a rank-matched received and trainable subspace, which can amplify update misalignment when resource budgets differ. By contrast, FEDHERA preserves a richer shared basis during the forward pass while updating only a budget-feasible prefix, thereby mitigating both local projection drift and cross-client directional drift.

### F.12. Profiling and Systems Overhead

FEDHERA decouples the downloaded rank from the trainable rank, allowing clients to use a richer global basis in the forward pass while optimizing only a budget-feasible prefix. A natural systems question is whether this decoupling substantially increases client-side resource consumption. To examine this, we profile a representative setting by comparing a coupled configuration, where $r^{\text{tot}} = r^{\text{train}}$, with a decoupled FEDHERA configuration, where the average downloaded rank is approximately $4\times$–$8\times$ larger than the trainable rank.

*Table 16.* **Profiling of coupled and decoupled rank configurations.**

| SETTING | CLIENT STEP LATENCY | PEAK GPU MEMORY | SERVER PIPELINE / ROUND |
|---|---|---|---|
| COUPLED ($r^{\text{tot}} = r^{\text{train}}$) | 737.3 MS | 8527 MB | 15.2 S |
| DECOUPLED ($r^{\text{tot}} > r^{\text{train}}$) | 763.1 MS | 8630 MB | 63.8 S |

Table 16 shows that increasing the downloaded rank introduces only modest client-side overhead: average training-step latency increases by $3.5\%$, and peak GPU memory increases by $1.2\%$. This is consistent with the design of FEDHERA: the

additional tail components are active in the forward pass but do not require optimizer states or gradient updates. Thus, the dominant client-side training cost remains governed by $r^{\mathrm{train}}$, rather than $r^{\mathrm{tot}}$.

The main additional overhead appears on the server side. The reported server time measures the end-to-end centralized pipeline, including update composition, per-layer SVD, rank allocation, adapter reconstruction, and redistribution. This cost increases from 15.2 s to 63.8 s per round in our reference implementation. Unlike client-side training, however, this server-side pipeline is centralized and more amenable to batching, parallelization, and stronger hardware support.

Overall, this profiling supports the systems motivation of FEDHERA. Rank decoupling converts otherwise unused download capacity into a richer shared forward basis, while keeping the client-side training footprint close to that of the coupled configuration. This enables the drift-resilience benefits of the frozen tail without substantially increasing the resource burden on heterogeneous clients.

