# OpenReview forum: "FedHera: Towards Drift-Resilient Federated Fine-tuning with Heterogeneous Resources"
_ICML.cc/2026/Conference — ICML 2026 regular_

### Official Review · Reviewer_21WY · 2026-02-22

**Soundness:** 3
**Presentation:** 3
**Significance:** 2
**Originality:** 2
**Overall Recommendation:** 4
**Confidence:** 3

**Summary:**

In a federated parameter-efficient fine-tuning setting, the communication capacity can be different from the computation capacity. For LoRA type of methods, the best download rank can be different (higher/lower) than the training rank. Enforcing them to be equal may create a bottleneck, which the authors propose FedHera to solve. Their goal is to better utilize the communication bandwidth to transmit a higher rank model, although not all parameters need to be trained due to computational limits. To choose how many rank components should be communicated and trained across layers, the paper proposes a water-filling algorithm to greedily choose the proportion of spectral energy to marginal cost. To avoid the noise of tail components in early training, they propose to modulate the effect of tail components considering the alignment of client update with global update direction and participation staleness. The extensive experiments and ablations show the superiority of the proposed method compared to the baselines.

**Compliance With Llm Reviewing Policy:**

Affirmed.

**Final Justification:**

The authors mostly resolved my concerns and stated that they will make clarifications in the paper. I kept my positive evaluation and score.

**Key Questions For Authors:**

1. Is the proposed "Layer-wise Spectrum-Preserving Water-Filling" strategy known in centralized (non-federated) literature? It seems applicable to broader literature and not a federated-specific solution to decide on the ranks of LoRA layers under some budget constraint. If so, can the authors discuss that while introducing the method?


2. Regarding the claim that the previous literature has coupled approaches, I remember a work from the last NeurIPS: Ravan: Multi-Head Low-Rank Adaptation for Federated Fine-Tuning, whose framework allows frozen parameters for some of the clients, although they are downloaded. It can be beneficial to discuss that previous literature has some decoupled approaches as well.


3. After Eq. 11, $\Delta W_i$ is defined as $\mathbf{B}_i^\text{up}\mathbf{A}_i^\text{up}$, but the update delta is defined in Eq. 10 as up$-$init parameters. It would be beneficial to use consistent notation.

**Limitations:**

Yes, they have. I don't think a further discussion on limitations and potential negative societal impact is needed.

**Strengths And Weaknesses:**

Strengths:

1. I have found the motivation interesting. The training/communication capacities can be different, and in such settings, we need more efficient algorithm designs considering this gap.

2. Paper is easy to read and well-presented.

3. Experimental evaluations are extensive, and ablation studies support the claims in the paper.


Weaknesses:

1. I think it is not well motivated that formulating rank allocation regarding the relative importances is a good idea. It is easy to see why choosing larger rank weights within the same layer is beneficial. However, it needs more explanation for the across-layer case. Let's say layer 1 has ranks [10, 9, 3, 2], and layer 2 has ranks [5, 5, 5, 5]. The algorithm with a budget of ~2 components (I know the cost is dynamic, but just take it as a simplified example) will prefer layer 1 twice. I am not sure proportions really show the importance comparison across layers. Shouldn't the weight magnitudes also be effective?

2. Since every client has different bandwidth and weight initialization at global rounds, it can cause a drift in the local training *across clients*, especially when the bandwidth heterogeneity increases. Although it sounds promising that every client receives a maximally informative update from global SVD, when there are high discrepancies across clients, I suspect local drift across clients, which can downgrade the performance. I see the ablation experiments of drift from ideal local updates. However, the drift I mention is the drift "across clients".

---

> ### Author Rebuttal · Authors · 2026-03-30
>
> We thank the reviewer for the constructive feedback.
> ## Q1.
>
> We agree that within-layer ordering is intuitive, while the motivation for cross-layer comparison can be made more explicit in the revision.
>
> Within each layer, the ordering is still determined by the singular spectrum itself. The role of the normalization in Eq. (4) is only to make components comparable across layers, where raw singular values are influenced by layer scale and parameterization. Accordingly, the allocator in Eq. (5) does not simply favor layers with larger absolute singular values; it allocates budget to components that retain more layer-relative energy per unit cost.
>
> Thus, absolute magnitude is not ignored: it still shapes the intra-layer spectrum and the benefit of retaining additional components. The normalization only prevents large-norm layers from dominating the global budget purely because of numerical scale.
>
> We also agree that budgeted rank allocation is not unique to federated learning. Related ideas appear in recent centralized SVD-based compression methods such as AdaSVD, SVD-LLM V2, and ASVD, which move beyond uniform truncation toward layer-specific allocation. What is federated-specific in FedHera is the way this principle is instantiated: the allocation is client-specific, jointly constrained by downlink and train-time memory budgets, subject to $r_{i,\ell}^{train} \le r_{i,\ell}^{tot}$, and coupled with the frozen-tail local optimization mechanism. We will clarify this distinction and connect our method more explicitly to the centralized literature.
>
> - [1] Adasvd: Adaptive singular value decomposition for large language models. arXiv:2502.01403
> - [2] Svd-llm v2: Optimizing singular value truncation for large language model compression. 10.18653/v1/2025.naacl-long.217
> - [3] Asvd: Activation-aware singular value decomposition for compressing large language models. arXiv:2312.05821
>
> ---
>
> ## Q2
>
> We thank the reviewer for pointing out Ravan, which is relevant and will be discussed in the revision. We view Ravan and FedHera as addressing different challenges in heterogeneous federated LoRA. Ravan [4] primarily improves adapter expressivity through a multi-head LoRA reparameterization, in which only the core matrices and lightweight scaling factors are trained.
> By contrast, FedHera is motivated by a different gap: in practical deployments, communication and local training resources are often asymmetric rather than tightly coupled. For SVD-truncated adapters, the rank a client can afford to download and use in the forward pass need not match the rank it can afford to train and upload.
>
> FedHera is built around this asymmetry. Rather than changing the adapter parameterization, it explicitly decouples the downloaded rank $r_{tot}$ from the trainable rank $r_{train}$ keeps the downloaded-but-untrained tail active in the forward pass as a stable spectral anchor, and progressively activates that tail via ATW. The goal is therefore not to enrich the LoRA form, but to reshape local optimization under heterogeneous communication/memory budgets and reduce truncation-induced drift. In this sense, Ravan studies adapter expressivity, whereas FedHera studies communication-training asymmetry in truncated heterogeneous FL. These directions are related but distinct, and we will revise the related work accordingly.
> - [4] Ravan: Multi-head low-rank adaptation for federated fine-tuning. arXiv:2506.05568
>
> We thank the reviewer for highlighting this complementary notion of drift. Our analysis focused on deviation from a high-rank reference update; here we additionally measure cross-client drift by reconstructing each participating client’s residual update in a common dense parameter space and computing round-wise pairwise directional dispersion via 1− cosine similarity.
>
> To address this point more thoroughly, we conducted additional experiments and added a complementary cross-client drift analysis. The new results show that, over rounds 0–9, FedHera consistently exhibits the lowest and most stable cross-client drift among the compared methods. This suggests that FedHera’s SVD-based server push not only reduces deviation from an ideal local update, but also provides a stronger shared anchor for heterogeneous clients, resulting in better-aligned local updates under bandwidth discrepancy. The added layer-wise analysis supports the same conclusion: under FedHera, drift remains relatively diffuse, whereas FlexLoRA and FedHL show sharper high-drift outlier layers, for example 0.7687 at `layers.18.self_attn.q_proj` for FlexLoRA and 0.5898 at `layers.6.self_attn.q_proj` for FedHL. We will include these new experimental results and analyses in the revision.

---

> > ### Author Rebuttal · Reviewer_21WY · 2026-04-02
> >
> > The authors have answered my questions and stated that they will make clarifications in the paper. I will keep my positive score.

---

### Official Review · Reviewer_sYzu · 2026-03-09

**Soundness:** 3
**Presentation:** 4
**Significance:** 3
**Originality:** 3
**Overall Recommendation:** 4
**Confidence:** 4

**Summary:**

The paper introduces a framework for federated finetuning (with LoRA) under resource heterogeneity. The main novelty is decoupling communication and computation ranks. This allows, for example, a client with high network bandwidth but limited ram to download a large modal. Methodology relies on assigning each client two sets of principal components of LoRA adapters for training and inference (can be higher rank). Experiments under skewed resource distributions are promising.

**Compliance With Llm Reviewing Policy:**

Affirmed.

**Key Questions For Authors:**

Questions:
- What is the impact on latency if r_total is much greeted than r_train?
- What is the impact of SVD (composition and reconstruction) overhead?

**Limitations:**

Yes

**Strengths And Weaknesses:**

Strengths:
- SVD-based aggregation is not novel but paper focuses on a well-motivated aspect of the federated finetuning problem, which is the potential decouple between communication and computation costs.
- Adaptive tail warm-up is interesting and novel. It helps to handle noisy updates at earlier communication rounds.
- Experiments are extensive (variety of models and datasets) and drift analysis is insightful.
- Paper is well-organized and written clearly.
- Related work discussion is satisfactory.

Weaknesses:
- Some details in section 4.4 can be introduced earlier (potentially in main figure) in more detail.
- Number of clients (100) in experiments is low and not reflective of practical cases.
- Detailed ablation on the analysis of design components (tail warm-up, prefix gating etc.) could be helpful.

---

> ### Author Rebuttal · Authors · 2026-03-30
>
> We thank the reviewer for the constructive feedback.
> ## Q1
>
> We measured this directly using a coupled setting $(r_{tot} = r_{train})$ and a decoupled one in which the average downloaded rank was 4x–8x the trainable rank.
>
> The average client train-step latency increased merely from 737.3 ms to 763.1 ms (+3.5%), and peak GPU memory increased from 8527 MB to 8630 MB (+1.2%). These results indicate that, although the frozen tail increases forward-pass cost in our representative setting, the overall client-side overhead remains modest even when $r_{\mathrm{tot}}$ substantially exceeds $r_{\mathrm{train}}$.
> In conclusion, the practical cost of decoupling is not dominated by client-side latency or memory growth.
>
> ---
>
> ## Q2
>
> We profiled the server composition + SVD + allocation + redistribution + push pipeline explicitly. In the same representative setting, the coupled configuration required 15.2 s per communication round on average, while the decoupled configuration required 63.8 s per round.
>
> Therefore, the main additional systems cost of decoupling is on the server side rather than on the client side. We note that this quantity measures the end-to-end server pipeline in our reference implementation, including composition, per-layer SVD, rank allocation, adapter reconstruction, and per-client serialization/redistribution. The larger increase in the decoupled setting is therefore expected, since larger r_tot affects the size of the redistributed client adapters and the associated server-side packaging/writing cost.
>
> By contrast, the client-side overhead remains small (+3.5% step latency and +1.2% peak memory), which is the key point for the frozen-tail concern.
>
> The absolute server times reported here should be interpreted as conservative estimates from an unoptimized research implementation. In deployment, this centralized server-side pipeline is substantially more amenable to hardware acceleration and systems optimization, such as stronger server hardware, batched linear algebra, and parallelized redistribution, than client-side training.
>
> ---
>
> ## Table 1
>
> | Setting | Avg client step latency | Avg peak GPU memory | Avg server SVD time / round |
> |---|---:|---:|---:|
> | Coupled $(r_{tot} = r_{train})$ | 737.3 ms | 8527 MB | 15.2 s |
> | Decoupled $(r_{tot} > r_{train}, avg. r_{tot} ≈ 4\times r_{train} \ to\ 8\times r_{train})$ | 763.1 ms | 8630 MB | 63.8 s |
>
> This shows that decoupling introduces only modest client-side overhead, while the main additional cost is the server-side SVD pipeline.
>
> In the revision, we will make two points more explicit. First, we will front-load key details of the ATW / prefix-gating mechanism currently introduced in Section 4.4, especially the early-round unreliability of the frozen tail and the server-side alignment score into the earlier overview for readability. Second, we will better surface the component-level evidence already provided in Section 6.2 and Appendix E, especially the stabilizing role of ATW in early rounds and the contribution of prefix-gated training.

---

> > ### Author Rebuttal · Reviewer_sYzu · 2026-04-03
> >
> > I thank authors for the rebuttal addressing my two questions about rank impact on latency and SVD overhead. My concerns are partially resolved and the weaknesses I mentioned regarding ablations and low number of clients remains. I keep my rating.

---

> > > ### Author Response · Authors · 2026-04-04
> > >
> > > We thank the reviewer again for the thoughtful follow-up.
> > >
> > > In the initial rebuttal, we first focused on providing measured evidence for the reviewer’s two explicit follow-up questions on latency and server-side overhead, while continuing to examine the remaining concerns on federation scale and component ablations. We have now completed these additional experiments and report them here.
> > >
> > > To address these remaining points more directly, we summarize the two additional targeted experiments below and will incorporate both the new evidence and the corresponding clarifications in the revision.
> > >
> > > First, regarding federation scale, our earlier addition broadened the evidence along the participation axis. To address the more direct concern that the original $N=100$ setting may be too limited, we further varied the total number of clients while keeping the number of participating clients per round fixed at 10, so that the experiment isolates the effect of a larger federation rather than simply aggregating more updates per round. On Qwen3-0.6B / E2E NLG for 20 rounds, FedHera remains the best method at $N=100/300/500$ on all four metrics, namely **held-out loss, ROUGE-L, cross-client drift**, and **absolute drift**. Its results are summarized below.
> > >
> > > | \(N\) | FedHera | FedHL | FlexLoRA | FedHeLLo |
> > > |---|---|---|---|---|
> > > | 100 | **0.5787 / 0.7899 / 0.4855 / 2.0601** | 0.6016 / 0.7731 / 0.5493 / 2.0724 | 0.6581 / 0.7745 / 0.6864 / 2.9912 | 0.7383 / 0.7663 / 0.6796 / 4.1703 |
> > > | 300 | **0.6075 / 0.7822 / 0.5780 / 2.1138** | 0.6350 / 0.7623 / 0.5829 / 2.1315 | 0.6610 / 0.7616 / 0.7092 / 3.1040 | 0.7464 / 0.7578 / 0.7221 / 4.1992 |
> > > | 500 | **0.6152 / 0.7710 / 0.6279 / 2.1455** | 0.6551 / 0.7615 / 0.6299 / 2.1868 | 0.7145 / 0.7585 / 0.7509 / 3.1588 | 0.7492 / 0.7572 / 0.7896 / 4.2087 |
> > >
> > > This broadens the evidence beyond the original $N=100$ regime and shows that the drift-resilience advantage persists as the federation becomes substantially larger and more fragmented.
> > >
> > > Second, regarding component-level validation, besides the ATW study already included in the paper, we added a more direct ablation on the frozen-tail / prefix-gated forward anchoring itself. We compare **full FedHera** against (i).**FedHera w/o ATW (static tail, $\lambda=1$)** and (ii).**FedHera w/o frozen-tail anchoring ($\lambda=0$, i.e., prefix-only forward)**. Importantly, the $\lambda=0$ variant still downloads the larger $r_{\text{tot}}$ subspace but does not use the frozen tail in the forward pass, so the extra downloaded components no longer provide optimization guidance and only add communication cost. Under this controlled ablation, full FedHera performs best, while removing ATW degrades performance and increases drift, and removing frozen-tail anchoring causes a much larger deterioration.
> > >
> > > | Variant | Final loss | ROUGE-L | Cross-client drift | Abs. drift |
> > > |---|---:|---:|---:|---:|
> > > | **Full FedHera** | **0.5787** | **0.7899** | **0.4855** | **2.0601** |
> > > | w/o ATW ($\lambda=1$) | 0.6016 | 0.7731 | 0.5577 | 2.1139 |
> > > | w/o tail anchoring ($\lambda=0$) | 0.8137 | 0.7398 | 0.7058 | 3.6325 |
> > >
> > > This directly supports that the gain is not merely from downloading a larger basis, but from using the frozen tail as a forward-pass anchor, with ATW further stabilizing when that anchor should become active.
> > >
> > > In the revision, we will also make this mechanism easier to follow by surfacing the intuition of ATW and prefix-gated forward anchoring earlier in the overview and main figure, so that the method description, ablation evidence, and drift analysis are connected more explicitly.
> > >
> > > Taken together, these additions directly address the two remaining weaknesses noted by the reviewer: they extend the empirical support beyond the original federation scale and more explicitly isolate the role of the key design components. We hope the reviewer finds that the remaining concerns are now substantially resolved and that the overall empirical support for the paper’s main claim is materially strengthened.

---

### Official Review · Reviewer_Rvba · 2026-03-12

**Soundness:** 2
**Presentation:** 2
**Significance:** 2
**Originality:** 2
**Overall Recommendation:** 3
**Confidence:** 4

**Summary:**

This paper presents FedHera, a heterogeneous federated LoRA fine-tuning framework that decouples downloaded and trainable ranks, using layer-wise rank allocation, prefix-gated updates, and a frozen tail to mitigate local drift and improve efficiency.

**Compliance With Llm Reviewing Policy:**

Affirmed.

**Final Justification:**

I thank the authors for the rebuttal. The response clarified several of my concerns and made the paper more convincing overall, although I still have some reservations about the empirical strength/novelty. I therefore raised my score from 2 to 3.

**Key Questions For Authors:**

Please refer to Weaknesses.

**Limitations:**

Yes.

**Strengths And Weaknesses:**

**Strengths**

[S1] The paper focuses on federated LoRA fine-tuning under heterogeneous client resources and highlights the mismatch between communication bandwidth and local training memory, which is a meaningful systems consideration.

[S2] The proposed framework is relatively well-structured. FedHera combines rank decoupling with layer-wise water-filling, prefix-gated training, and a frozen-tail mechanism, forming a coherent design for handling heterogeneous resource constraints.

**Weaknesses**

[W1] Baseline coverage is not sufficiently comprehensive. Although the related-work section explicitly discusses closely related heterogeneous LoRA methods such as FLoRA and Heterogeneous LoRA, the main experiments only compare against FedHomoLoRA, FlexLoRA, FedHL, and FedHeLLo. This makes the empirical evidence less convincing, and stronger comparisons against directly relevant federated LoRA baselines (e.g., FLoRA [1], FFA-LoRA [2], HetLoRA [3], and FedSA-LoRA [4], when comparable) would strengthen the claims.

[W2] The methodological novelty appears somewhat incremental. While the dual-rank decoupling idea is reasonable, the overall framework mainly combines existing design elements from prior federated LoRA work, including SVD-based rank adaptation, prefix masking/freezing, and residual aggregation, which limits the degree of conceptual novelty.

[W3] The empirical setup may be too limited to fully support the “drift-resilient” claim. The appendix reports results with $N=100$ clients, $C=0.1$ participation, and only $R=10$ communication rounds, averaged over three runs. For a paper centered on accumulated local drift, it remains unclear whether the same conclusions would hold under different federation scales (e.g., 30 or 50 clients), higher participation rates, or longer training horizons.

[W4] Several key design choices are heuristic and not fully validated. The water-filling procedure depends on per-layer time/memory slopes and adaptive scalarization weights, while the frozen-tail mechanism still performs the forward pass in the full rtot-rank subspace. The paper does not sufficiently clarify how these costs are profiled in practice, how stable they are across hardware settings, or how sensitive performance is to these design choices.

*References*

[1] Wang, Z., Shen, Z., He, Y., Sun, G., Wang, H., Lyu, L., & Li, A. (2024). Flora: Federated fine-tuning large language models with heterogeneous low-rank adaptations. Advances in Neural Information Processing Systems, 37, 22513-22533.

[2] Sun, Y., Li, Z., Li, Y., & Ding, B. (2024). Improving lora in privacy-preserving federated learning. arXiv preprint arXiv:2403.12313.

[3] Cho, Y. J., Liu, L., Xu, Z., Fahrezi, A., & Joshi, G. (2024, November). Heterogeneous lora for federated fine-tuning of on-device foundation models. In Proceedings of the 2024 conference on empirical methods in natural language processing (pp. 12903-12913).

[4] Guo, P., Zeng, S., Wang, Y., Fan, H., Wang, F., & Qu, L. (2024). Selective aggregation for low-rank adaptation in federated learning. arXiv preprint arXiv:2410.01463.

---

> ### Author Rebuttal · Authors · 2026-03-30
>
> We thank the reviewer for the constructive feedback. FedHera’s core contribution is to decouple reception and trainable ranks, together with forward-active frozen tails and ATW, to mitigate truncation-induced local drift under heterogeneous communication and memory budgets.
>
> ## W1.
> Our original baselines were chosen to cover complementary forms of heterogeneous federated fine-tuning: homogeneous bottlenecking (FedHomoLoRA), coupled SVD-based heterogeneity (FlexLoRA), server-side residual debiasing (FedHL), and pruning-style heterogeneity (FedHeLLo). To add a more direct comparison, we further evaluated FLoRA and HetLoRA on the same backbone. Across Alpaca and E2E NLG, under both Uniform(A) and Skewd(B) resource distribution, FedHera remains best on average:
>
> - Alpaca-A 1.121/0.708 vs. 1.195/0.703 vs. 1.212/0.697
> - Alpaca-B 1.140/0.709 vs. 1.202/0.698 vs. 1.222/0.692
> - E2E-A 0.490/0.879 vs. 0.512/0.860 vs. 0.791/0.758
> - E2E-B 0.492/0.875 vs. 0.553/0.838 vs. 0.880/0.716
>
> (FedHera / FLoRA / HetLoRA in Loss / ROUGE-L).
>
> The gap is especially clear in skewed settings, where coupled-rank methods force low-resource clients into more severely truncated subspaces. These additions therefore strengthen rather than change our conclusion. FFA-LoRA and FedSA-LoRA are also relevant and will be added to the related work, although they are not equally direct baselines due to their different objectives and assumptions.
>
> ---
> ## W2.
> We agree the novelty boundary should be stated more sharply. We do not claim novelty in SVD redistribution or heterogeneous ranks by themselves. Rather, FedHera’s novelty lies in their combination around a resource-decoupled formulation: separating downloaded rank $r_{tot}$ from trainable rank $r_{train}$, using a forward-active frozen tail as a spectral anchor, progressively activating it via ATW, and doing so in a client-side mechanism compatible with either standard or residual server aggregation. This distinguishes FedHera from related methods: FLoRA and HetLoRA support heterogeneous ranks but do not decouple reception and trainable ranks or use inactive downloaded capacity as a forward anchor, while FedHL addresses drift mainly on the server side. We will sharpen this distinction in the revision.
>
> ---
>
> ## W3.
> The appendix drift study was intended as a controlled mechanism analysis under a fixed federation regime, not an exhaustive scale sweep. Its purpose was to isolate truncation-induced local drift under heterogeneous budgets, where FedHera already shows lower deviation from the high-rank reference. To broaden this evidence, we added 20-round experiments at higher client selection rates ($C=0.2, 0.3$) on the generative-task setting, comparing FedHera with FedHL and FLoRA, and also included a complementary cross-client drift metric based on the directional dispersion of reconstructed dense residual updates.
>
> Across these broader settings, FedHera continues to show the same advantage as in the paper. Notably, as participation increases, its cross-client drift does not increase and even decreases slightly at the same horizon (Dispersion = 0.413049 at $C=0.2$ vs. 0.373893 at $C=0.3$), while remaining better than the baselines. This is consistent with the same mechanism-level interpretation: decoupling downloaded and trainable ranks preserves a richer global anchor, reducing local drift and improving cross-client alignment. FedHera mitigates truncation-induced drift under heterogeneous resource budgets, and the added experiments broaden the evidence beyond the original controlled setting.
>
> ---
> ## W4.
> We agree the profiling-and-mapping protocol should be stated more explicitly. FedHera does not rely on hand-tuned, device-specific latency constants. The allocator uses profiled marginal per-rank costs for download, memory, and step time, obtained from a short bootstrap phase by sweeping a small set of ranks and fitting $c_{i,\ell}^{mem}$ and $c_{i,\ell}^{time}$. Thus Eq. (7) and Algorithm 2 use measured costs rather than fixed manual constants. In addition, $\alpha,\beta$ are recomputed from residual time and memory budgets at each allocation step rather than manually tuned.
>
> To address overhead directly, we profiled a representative Alpaca/Qwen3-0.6B setting, comparing coupled ($r_{tot}=r_{train}$) and decoupled ($r_{tot}>r_{train}, avg. r_{tot}\approx 4\times  r_{train} \ to \  8\times r_{train}$) configurations. Decoupling increases average client step latency from 737.3 ms to 763.1 ms (+3.5%) and peak GPU memory from 8527 MB to 8630 MB (+1.2%), while the main extra cost appears on the server side, where composition + SVD + allocation + redistribution + push increases from 15.2 s to 63.8 s per round. Thus, client-side overhead remains modest, and most additional cost comes from the server-side SVD pipeline. We will make both the profiling protocol and the cost breakdown explicit in the revision. Please refer to **Table 1** in our response to **Reviewer sYzu** for detailed results.

---

> > ### Author Rebuttal · Reviewer_Rvba · 2026-04-02
> >
> > Thanks for the detailed rebuttal. My main concerns have been partially addressed, and I am updating my score to 3.

---

> > > ### Author Response · Authors · 2026-04-06
> > >
> > > We thank the reviewer again for the careful follow-up and for updating the score. We understand that the remaining concerns are mainly about two points: whether the empirical support for the drift-related claim is broad enough, and whether the paper makes the methodological distinction from prior heterogeneous LoRA approaches sufficiently clear.
> > >
> > > To address the first point more directly, we now have additional results on larger federation scales. Beyond the original $N=100$ setting, we further varied the total number of clients while keeping the number of participating clients per round fixed at 10, so that the comparison isolates the effect of a larger and more fragmented federation rather than simply aggregating more client updates in each round. Since the reviewer also asked for stronger positioning against closely related heterogeneous LoRA methods, we additionally include FLoRA in this comparison. On Qwen3-0.6B / E2E NLG over 20 rounds, FedHera remains the best method at $N=100/300/500$ on all four tracked quantities. The results are summarized below, where each entry is reported as loss / ROUGE-L / cross-client drift / absolute drift.
> > >
> > > | \(N\) | FedHera | FedHL | FlexLoRA | FLoRA | FedHeLLo |
> > > |---|---|---|---|---|---|
> > > | 100 | **0.5787 / 0.7899 / 0.4855 / 2.0601** | 0.6016 / 0.7731 / 0.5493 / 2.0724 | 0.6581 / 0.7745 / 0.6864 / 2.9912 | 0.6416 / 0.7726 / 0.6398 / 2.4160 | 0.7383 / 0.7663 / 0.6796 / 4.1703 |
> > > | 300 | **0.6075 / 0.7822 / 0.5780 / 2.1138** | 0.6350 / 0.7623 / 0.5829 / 2.1315 | 0.6610 / 0.7616 / 0.7092 / 3.1040 | 0.6663 / 0.7656 / 0.6526 / 2.7088 | 0.7464 / 0.7578 / 0.7221 / 4.1992 |
> > > | 500 | **0.6152 / 0.7710 / 0.6279 / 2.1455** | 0.6551 / 0.7615 / 0.6299 / 2.1868 | 0.7145 / 0.7585 / 0.7509 / 3.1588 | 0.6838 / 0.7649 / 0.6945 / 2.8146 | 0.7492 / 0.7572 / 0.7896 / 4.2087 |
> > >
> > > We believe this broadens the evidence beyond the original regime and also strengthens the direct comparison to a reviewer-suggested heterogeneous LoRA baseline. Importantly, the advantage persists not only in held-out quality but also in both drift measures as the federation becomes substantially larger and more fragmented.
> > >
> > > To address the second point more directly, we now also have additional mechanism-level results that make the source of the gain clearer. Besides the ATW study already included in the paper, we further compare full FedHera against a static-tail variant without ATW with $\lambda=1$, and against a prefix-only forward variant without frozen-tail anchoring with $\lambda=0$. Under this controlled comparison, full FedHera performs best, removing ATW causes a clear degradation, and removing frozen-tail anchoring causes a much larger deterioration in both performance and drift. We believe these results make the methodological distinction more concrete, because the improvement does not come simply from downloading a larger basis, but from the resource-decoupled formulation together with the forward-active frozen tail, while ATW further stabilizes when that guidance should become active.
> > >
> > > This is also how we intend to sharpen the novelty boundary in the revision. Our claim is not that SVD redistribution, rank heterogeneity, freezing, or residual aggregation are individually new. Rather, the contribution of FedHera is the explicit separation between reception rank and trainable rank, together with the client-side optimization mechanism that this separation enables. The additional mechanism-level results help make this point more explicit, because they show that the gain is tied to how the extra received subspace is used during optimization, rather than to larger downloaded capacity alone.
> > >
> > > On baseline coverage, we also appreciate the reviewer’s earlier suggestions. In the rebuttal, we already added direct comparisons to FLoRA and HetLoRA. In the revision, we will incorporate these comparisons into the paper more explicitly. We will also expand the related work discussion to cover FFA-LoRA and FedSA-LoRA more clearly, so that the positioning relative to adjacent LoRA-based FL methods is easier to follow.
> > >
> > > Taken together, we hope the reviewer finds that the remaining concerns are now addressed more directly. The additional federation-scale results broaden the empirical support for the drift-related claim and further strengthen the comparison to closely related heterogeneous LoRA baselines, while the additional mechanism-level results make the key methodological contribution easier to isolate and interpret. We therefore hope the reviewer will find that the overall support for the paper’s main claim is now substantially strengthened.

---

### Official Review · Reviewer_y4dk · 2026-03-13

**Soundness:** 3
**Presentation:** 3
**Significance:** 3
**Originality:** 3
**Overall Recommendation:** 5
**Confidence:** 4

**Summary:**

This paper proposes FedHera, a federated fine-tuning framework for large language models that addresses resource heterogeneity by decoupling the downloaded adapter rank from the locally trainable rank, enabling clients to leverage richer global information while maintaining efficient local training.

**Compliance With Llm Reviewing Policy:**

Affirmed.

**Final Justification:**

The authors have addressed my concerns, and I will keep my positive score.

**Key Questions For Authors:**

1. Results on additional naive baselines are missing, which would help better illustrate the advantages of the proposed method and provide a clearer reference point for performance gains.

Specifically, while the authors include a FedHomoLoRA baseline where all clients are constrained to the minimum rank budget, several straightforward baselines are absent. For example, a setting in which all clients perform full-rank fine-tuning could serve as an upper bound, and another where resource-constrained clients abstain from training would reflect a practical alternative. Including such baselines would strengthen the empirical evaluation.

2. Some relevant prior work may be further discussed. For example, Fed-PLoRA [1] also explores leveraging global information (sacrificing some of the communication efficiency) while maintaining efficient local fine-tuning. Discussing its relationship to the proposed method would help clarify the differences and contributions.

[1] Heterogeneous Federated Fine-Tuning with Parallel One-Rank Adaptation.

**Limitations:**

Not mentioned.

**Strengths And Weaknesses:**

1. The paper addresses an important problem in federated fine-tuning of large language models under heterogeneous client resources, which is highly relevant for practical deployments.

2. The proposed framework is well-motivated, introducing a novel resource decoupling design that separates the downloaded rank from the trainable rank, and provides an intuitive and practical way to better utilize available communication bandwidth.

3. The introduced Unbiased Aggregation eliminates the bias introduced by heterogeneous rank truncation by preserving truncated global components and aggregating only valid client residuals.

4. The convergence analysis is rigorous, clearly showing how adaptive rank allocation reduces approximation error and leads to tighter generalization bounds.

5. The paper presents extensive experiments across multiple reasoning and generation benchmarks, showing consistent improvements over several heterogeneous federated fine-tuning baselines.

6. For weaknesses, please refer to the questions.

---

> ### Author Rebuttal · Authors · 2026-03-30
>
> We thank the reviewer for the constructive feedback.
> ## Q1
> We agree that two additional reference points are particularly useful here:
> 1. a simple practical abstention baseline, where the most resource-constrained clients do not perform local training, and
> 2. an unconstrained upper-bound reference that indicates what performance would look like without the tight low-rank training constraint.
>
> To address this, we added a targeted experiment in the skewed resource setting on our main generative backbone, Llama-3.2-3B, using two representative tasks. For the abstention-style reference, we extend the original client tiers by including a lowest tier with $r_{train} = 0$, i.e., these clients download the model but do not contribute local updates. For the upper-bound side, rather than introducing a full full-parameter federated fine-tuning protocol, which falls outside the resource-constrained setting studied in this paper, we report a high-rank oracle-like reference as a ceiling comparison in the same experimental regime.
> The results are informative:
> - **Alpaca:** abstention baseline 1.881 / 0.603, FedHera 1.140 / 0.709, high-rank oracle-like reference 1.089 / 0.721
> - **E2E NLG:** abstention baseline 2.026 / 0.672, FedHera 0.493 / 0.875, high-rank oracle-like reference 0.459 / 0.894
>
> (Each pair reports Loss / ROUGE-L).
>
> These results support two points. First, simply letting the weakest clients abstain from training is a substantially weaker practical alternative than FedHera, especially in the skewed regime. Second, FedHera remains much closer to the high-rank oracle-like reference than to the abstention baseline, while still operating fully within the intended heterogeneous resource-constrained setting. We will incorporate this clarification in the revision to make the practical positioning of FedHera more explicit.
>
> ---
> ## Q2
> We thank the reviewer for pointing us to Fed-PLoRA, which is relevant and will be added to the revised related-work discussion. Our understanding is that Fed-PLoRA addresses heterogeneous federated LoRA through a different adapter construction from FedHera. In particular, Fed-PLoRA replaces the classical multi-rank LoRA module with multiple parallel one-rank modules and uses a Select-N-Fold strategy so that each client trains only a subset of them according to its resource budget, with the stated goal of reducing initialization and aggregation noise under heterogeneity.
>
> By contrast, FedHera does not redesign the LoRA parameterization itself. Instead, it studies heterogeneity within the standard multi-rank / truncated-rank LoRA formulation, and focuses on a different gap: In practice, the rank a client can support for forward computation may differ from the rank it can train and transmit back. FedHera therefore explicitly decouples $r_{tot}$ and $r_{train}$, keeps the downloaded-but-untrained tail active in the forward pass, and progressively activates it via ATW to mitigate truncation-induced drift.
>
> In this sense, Fed-PLoRA and FedHera are related, but they operate at different levels of the design space: Fed-PLoRA changes the adapter construction itself, whereas FedHera changes the resource formulation and optimization behavior under heterogeneous budgets while remaining compatible with the standard LoRA form. We will revise the related-work section to make this boundary clearer.

---

> > ### Author Rebuttal · Reviewer_y4dk · 2026-03-31
> >
> > The authors have addressed my concerns, and I will keep my positive score.

---

### Decision · Program_Chairs · 2026-04-30

**Decision:**

Accept (regular)

**Comment:**

This paper received positive while slightly divergent scores (3, 4, 4, 5). Two of the positive reviews are  relatively short. Reviewers found this paper easy to read with interesting motivations, and extensive experiments that are further strengthened during rebuttal. Reviewers also have remaining concerns on novelty and empirical results, and the reviewer also acknowledged the novelty question is not easily addressed during rebuttal. Overall, because of the reviewers’s interests in the paper and there are no obvious flaws in the paper, the incremental contributions of the paper would be valuable to present to the community.